# Assessment of PRMT6-dependent alternative splicing in pluripotent and differentiating NT2/D1 cells

Matthias Eudenbach[1], Jonas Busam[2,4], Caroline Bouchard[1], Oliver Rossbach[3], Kathi Zarnack[2,4], Uta-Maria Bauer[1]

Protein arginine methyltransferase 6 (PRMT6) is a well-characterized epigenetic regulator that methylates histone H3 at arginine 2 (H3R2me2a) in both promoter and enhancer regions, thereby modulating transcriptional initiation. We report here that PRMT6 also regulates gene expression at the post-transcriptional level in the neural pluripotent state and during neuronal differentiation of NT2/D1 cells. PRMT6 knockout causes widespread alternative splicing changes in NT2/D1 cells, most frequently cassette exon alterations. Most of the PRMT6-dependent splicing targets are not transcriptionally affected by the enzyme and regulated in an H3R2me2a-independent manner. However, for a small subset of splicing events, the PRMT6-mediated deposition of H3R2me2a overlaps with the splice site, suggesting a potential dual function in both transcriptional and co-/post-transcriptional regulation. The splicing targets of PRMT6 include ribosomal proteins, splicing factors, and chromatin-modifying enzymes such as PRMT4, DNMT3B, and ASH2L, some of which are associated with differentiation decisions. Taken together, our results in NT2/D1 cells show that PRMT6 exerts predominantly H3R2me2a-independent functions in RNA splicing, which may contribute to pluripotency and neuronal identity.

## Introduction

Protein arginine methyltransferase 6 (PRMT6) belongs to the type I PRMT subfamily that asymmetrically dimethylates arginine residues in a wide range of substrate proteins (Rakow et al, 2020; Hamey et al, 2021). The enzyme is predominantly localized in the nucleus and, similar to other PRMT members, involved in the epigenetic regulation of gene expression (Frankel et al, 2002; Hyllus et al, 2007). In addition, PRMT6 participates in RNA splicing, DNA repair, and signal transduction (Gupta et al, 2021). It consequently contributes to several key biological processes, such as development and differentiation, proliferation, senescence, genome stability, and cell–cell communication.

To execute gene-regulatory functions, PRMT6 is recruited to the chromatin by transcription factors (Harrison et al, 2010; Herglotz et al, 2013; Herkt et al, 2018; Hwang et al, 2019; Gerstner et al, 2024). PRMT6 mediates both transcriptional activation and repression through methylation of various nuclear substrate proteins, including histone H3 at arginine 2 (H3R2me2a). H3R2me2a counteracts the deposition of active promoter marks, such as the neighboring histone H3 lysine 4 trimethylation (H3K4me3) mark. The H3R2me2a-dependent repulsion and catalytic inhibition of the H3K4me3-executing KMT2A co-activator complex lead to decreased transcriptional initiation (Guccione et al, 2007; Hyllus et al, 2007; Iberg et al, 2008; Stein et al, 2012; Bouchard et al, 2018). The H3R2 residue is also targeted by PRMT5-catalyzed symmetrical dimethylation (H3R2me2s), which is an active histone mark (Migliori et al, 2012). These mutually exclusive promoter-associated H3R2 modifications have opposite transcriptional functions. Consistent with its gene-repressive function, PRMT6 cooperates with Polycomb-repressive complexes (PRC) and enhances the PRC2-mediated deposition of H3K27me3 (Herglotz et al, 2013; Kuvardina et al, 2015; Stein et al, 2016). At enhancers, PRMT6-deposited H3R2me2a is recognized by a subset of PHDs of the KMT2D protein, the major H3K4 mono-methyltransferase. This reading of the epigenetic code facilitates chromatin recruitment of the KMT2D co-activator complex and deposition of the key activating enhancer marks H3K4me1 and H3K27ac, followed by transcriptional activation of the corresponding genes (Bouchard et al, 2018).

Although it is well established that PRMT6 orchestrates the regulation of transcriptional initiation through histone arginine methylation and by modulating the deposition of adjacent histone marks, its role in the post-transcriptional steps of gene expression and RNA processing is far less understood. PRMT6 has been reported to simultaneously regulate transcriptional initiation and pre-mRNA splicing of a subset of target genes in human breast cancer cells (Harrison et al, 2010; Dowhan et al, 2012). Furthermore, the splicing-related functions of PELP1 and PTEN were found to be influenced by PRMT6 (Mann et al, 2014; Feng et al, 2019). Profiling of the cellular interactome and arginine methylome of several PRMT members, including PRMT6, uncovered a substantial amount of

[1]Institute for Molecular Biology and Tumor Research (IMT), Philipps-University Marburg, Marburg, Germany   [2]Buchmann Institute for Molecular Life Sciences (BMLS) and Institute of Molecular Biosciences, Goethe University Frankfurt, Frankfurt, Germany   [3]Institute of Biochemistry, Faculty of Biology and Chemistry (FB08), Justus-Liebig-University of Giessen, Giessen, Germany   [4]Theodor Boveri Institute, Biocenter, University of Würzburg, Würzburg, Germany

Correspondence: bauer@imt.uni-marburg.de

PRMT6-mediated methylation in RNA-binding proteins (RBPs) such as splicing factors (SFs) (Wei et al, 2021). Specifically, the RG/RGG-repeat domains, known to be essential for the functions of many RBPs in splicing, localization, and stability of mRNAs, were identified to be arginine-methylated, in agreement with the substrate preference of PRMT6 (Hamey et al, 2021). However, PRMT6 has not been shown to play a direct role in regulating pre-mRNA processing.

Our previous work revealed that PRMT6 is an important regulator of the gene expression program in human NT2/D1 cells through promoter- and enhancer-associated methylation of H3R2 (Hyllus et al, 2007; Stein et al, 2016; Bouchard et al, 2018). NT2/D1 cells resemble neural stem cells in the developing fetal central nervous system (CNS) and differentiate into neuronal cells upon treatment with all-trans retinoic acid (ATRA). By means of transcriptional functions, PRMT6 contributes to both neural pluripotency and neuronal differentiation of NT2/D1 cells and of neural progenitor cells in the murine fetal CNS (Bouchard et al, 2018). Given that (i) a putative contribution of PRMT to alternative splicing (AS) in the context of neuronal differentiation has not yet been studied and (ii) AS of mRNAs is a fundamental process by which many cell types, also cells in the CNS, expand their transcriptome diversity (Raj & Blencowe, 2015), we investigated the role of PRMT6 in mRNA processing using RNA-seq data sets of PRMT6 WT (control, CT) and knockout (KO) NT2/D1 cells (Bouchard et al, 2018). We employed untreated and ATRA-treated NT2/D1 cells to determine the PRMT6-dependent splicing function in pluripotent and differentiating neuronal cells. We distinguished transcriptional and splicing effects and determined the co-occurrence of H3R2me2a at PRMT6-regulated splice sites by implementing ChIP-seq data sets of H3R2me2a (Bouchard et al, 2018). Our results show that PRMT6 globally affects AS in NT2/D1 cells, primarily through splicing of cassette exons and mainly in a histone methylation–independent manner, potentially contributing to pluripotency and neuronal identity.

# Results

### PRMT6 impacts alternative splicing in pluripotent NT2/D1 cells, with cassette exons being the most common PRMT6-regulated splicing alterations

Splicing is predominantly a co-transcriptional process, known to be influenced by chromatin and transcriptional regulators (Luco et al, 2010). PRMT6 functions as a transcriptional co-regulator and impacts transcriptional initiation in a position-dependent manner, either positively or negatively, for example, in differentiating neuronal cells (Hyllus et al, 2007; Stein et al, 2016; Bouchard et al, 2018). To investigate its putative role in regulating co-transcriptional steps of RNA maturation, such as splicing, we used mRNA transcriptome sequencing (RNA-seq) of NT2/D1 cell lines containing either WT (control, CT) or genetically inactivated (knockout, KO) PRMT6 gene loci (Figs 1A and S1A). NT2/D1 KO cells displayed a complete loss of PRMT6 expression, as well as a global reduction of arginine-methylated proteins in comparison with CT cells (Fig S1B, Bouchard et al, 2018).

To dissect the PRMT6-dependent splicing pattern, we first focused on undifferentiated pluripotent NT2/D1 cells (in the absence of ATRA, −) and quantified the local splicing variations (LSVs) upon PRMT6 knockout using MAJIQ (Vaquero-Garcia et al, 2016). Using thresholds for splicing changes of ΔPSI ≥ 0.05 with a probability ≥ 0.9, we identified 233 alternative splicing (AS) events that occurred within 173 genes upon PRMT6 knockout in NT2/D1 cells (Fig 1B, Table S1). The PRMT6-regulated AS events were classified into the following five major categories: cassette exons (i.e., exons included or skipped during splicing), intron retention (i.e., introns removed or kept during splicing), alternative 3′ splicing sites (i.e., exons extended at their beginning), alternative 5′ splicing sites (exons shortened at their end), and finally the combination of alternative 3′ and 5′ splicing sites in the same exon. We found that cassette exons were the most prevalent and accounted for more than half (67.4%) of the PRMT6-regulated AS events in KO cells (Fig 1C).

Among the top 50 PRMT6-regulated splicing targets, including cassette exons and mutually exclusive exons, exon exclusions occurred most frequently at 50% with the PRMT6 knockout, followed by exon inclusions at 30% and mutually exclusive exons at 20%. (Fig S2). For validation, we selected representative gene transcripts of the major PRMT6-regulated splicing categories (i.e., cassette exons and mutually exclusive exons), including splicing events with high ΔPSI values. To this end, we subjected NT2/D1 CT and KO cells to semiquantitative/standard RT–PCR analysis for the corresponding transcript isoforms of *PRMT4*, *CHASERR*, *FLNB*, and *CALD1* (Figs 1D–I and S3). Thereby, we confirmed that PRMT6 knockout causes significant splicing changes in NT2/D1 cells, that is, increased exon exclusion in *PRMT4* exon 15 (E15) (Fig 1D and G) and *CHASERR* E2 (Fig 1E and H), increased exon inclusion in *FLNB* E30 (Fig 1F and I), and mutually exclusive exons E3a/3b of *CALD1* (Fig S3). Although the standard RT–PCR results of some AS events did not fully replicate the splicing extent determined by the RNA-seq data (e.g., in case of *CHASERR* and *FLNB*), likely because of different molecular biases of the two methodologies, the PRMT6-mediated effects and tendencies were reproduced by our PCR validations, as also verified by the good correlation of RNA-seq–derived and standard RT–PCR–derived ΔPSI values in scatter plot analysis (Fig S4).

Interestingly, the E15-containing full-length (FL) *PRMT4* transcript isoform has been reported as the major transcript isoform in the brain tissue and exclusion of E15 results in the loss of the automethylation site R551, thereby impinging on PRMT4's transcriptional and splicing functions (Kuhn et al, 2011; Wang et al, 2013). To examine whether these PRMT6-regulated AS events are cell line–dependent effects, for example, specific for NT2/D1 cells, we generated HeLa, HEK293T, and U2OS cell lines containing a PRMT6 knockout (KO) by means of CRISPR/Cas9 genome editing (Fig S5A). Standard RT–PCR analysis of the corresponding short and long isoforms of *PRMT4* and *FLNB* indicated that the three cell lines in their PRMT6 WT (CT) state display a more or less predominant expression of the *PRMT4* ΔE15 and FLNB ΔE2 isoforms (Fig S5B) in contrast to NT2/D1 CT cells, in which the FL isoforms prevail for both transcripts, most clearly for *PRMT4* (Fig 1D and F). PRMT6 deficiency (KO) did not affect the *PRMT4* isoform ratio in HeLa cells compared with CT cells, but caused decreased ΔE15 levels in HEK293T cells and increased ΔE15 levels in U2OS cells, the latter effect being similar to the PRMT6-mediated regulation in NT2/D1 cells. PRMT6 deletion did

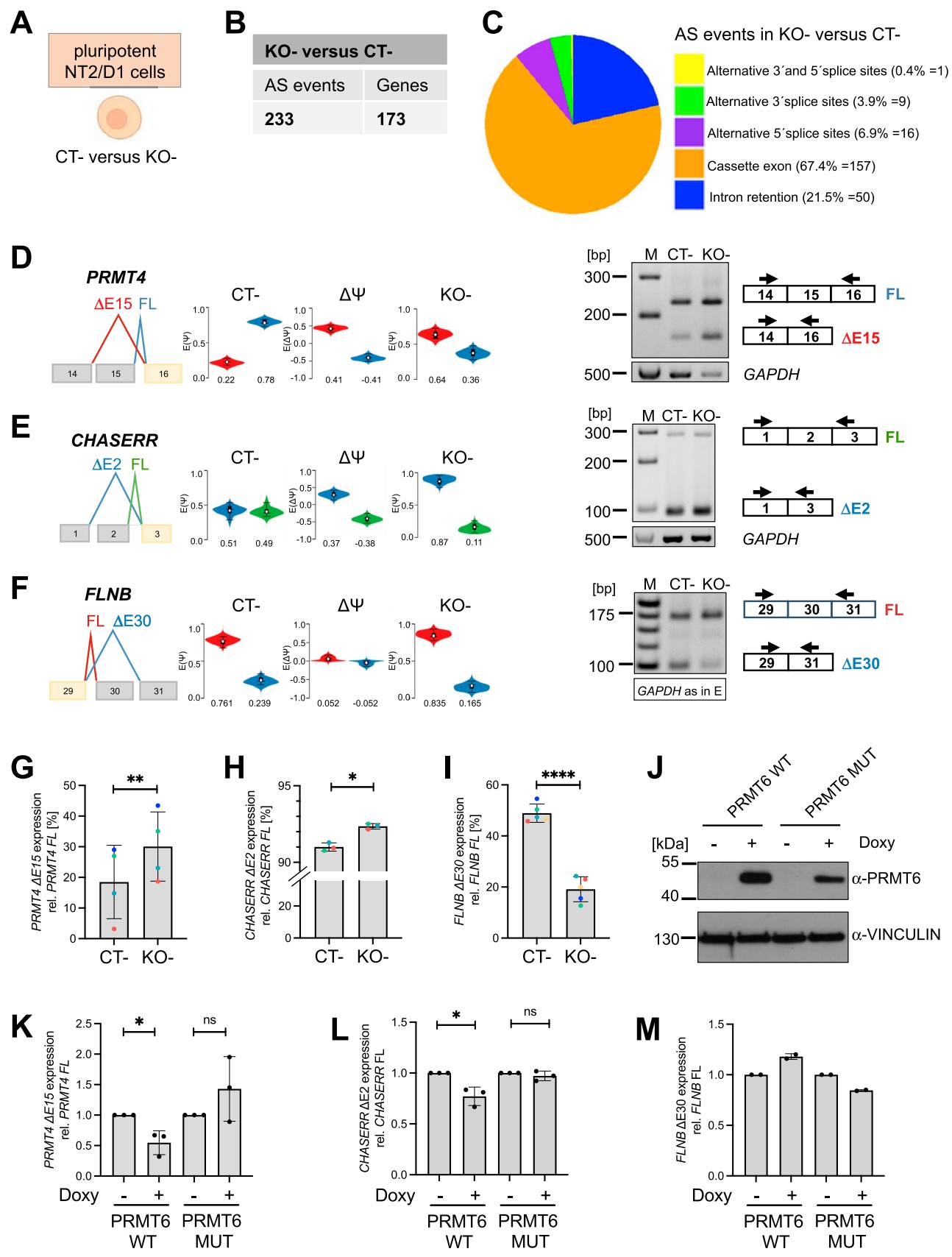

not alter the *FLNB* isoform expression in the three tested cell lines (Fig S5B). These findings suggest that the AS targets of PRMT6, as examined here for two examples, are regulated in a cell line–dependent manner.

To examine whether AS regulation by PRMT6 is dependent on its enzymatic activity, we performed rescue experiments in NT2/D1 KO cells. To this end, we re-expressed WT or enzymatically inactive mutant (MUT) PRMT6 in a doxycycline-inducible manner in KO cells, as verified at the RNA and protein levels (Figs S6 and 1J). WT PRMT6, but not the mutant, reversed the changes in AS in *PRMT4*, *CHASERR*, and *FLNB* caused by PRMT6 deletion and significantly reduced the level of the exon-excluding transcript isoforms of *PRMT4* and *CHASERR* (Fig 1K–M). Taken together, these results show that PRMT6 regulates gene expression through AS in a cell type–dependent manner, predominantly at the level of alternative cassette exons, in a widespread and methyltransferase-dependent manner in NT2/D1 cells.

## Splicing targets are predominantly regulated in a histone modification– and transcription-independent manner by PRMT6

Using gene ontology (GO) analysis, we next searched for biological processes and pathways enriched in the splicing targets of PRMT6 in pluripotent NT2/D1 cells (Fig 2A). The overrepresented GO terms included translation, co-translational protein targeting, rRNA processing, ribosome biogenesis, RNA processing, and gene expression, indicating that the enzyme affects essential cellular processes by its splicing regulatory function, in agreement with the growth defects observed in NT2/D1 cells upon PRMT6 knockout (Bouchard et al, 2018). As splicing outcomes can be regulated by characteristic histone modification patterns enriched at nucleosomes associated with the splice sites (Luco et al, 2010; Agirre et al, 2021), we questioned whether PRMT6 might directly influence AS by H3R2me2a deposition and/or modulating the occurrence of adjacent histone modifications. Recently defined splicing-associated chromatin signatures (SACS) include, among other histone modifications, H3K4me1, H3K4me3, and H3K27ac (Sims et al, 2007; Agirre et al, 2021; Segelle et al, 2022),

which are known to be influenced by the PRMT6-catalyzed H3R2me2a (Hyllus et al, 2007; Stein et al, 2016; Bouchard et al, 2018). To investigate the potential direct impact of PRMT6 and its predominant histone mark H3R2me2a on AS, we compared the PRMT6-regulated splicing events identified here with the PRMT6-dependent H3R2me2a occurrence from ChIP-seq data sets of NT2/D1 CT cells (Bouchard et al, 2018). Thereby, 80% of the alternatively spliced genes (139 of 173 total splicing targets, Fig 2B) displayed either H3R2me2a enrichment peaks not overlapping with the splice site (e.g., *PRMT4*, Fig 2C) or even no H3R2me2a deposition above background (IgG ChIP control enrichment) within the target gene locus (e.g., *CHASERR*, Fig 2D), suggesting that in most of the cases, PRMT6 affects splicing independently of H3R2me2a in NT2/D1 cells. Similarly, other histone modifications, such as H3K4me1, H3K4me3, and H3K27ac, were either not enriched at the splice site of these genes or did not show splicing event–dependent alterations in their abundance in NT2/D1 KO cells (Fig 2C and D). However, at 20% of the PRMT6-regulated splicing targets (34 of 173 total splicing targets, Fig 2B), H3R2me2a was deposited at the relevant splice sites, as exemplified for three ribosomal protein genes (Fig S7A–C). Despite this overlap, the SACS-related, H3R2me2a-adjacent histone marks H3K4me1, H3K4me3, and H3K27ac did not show any dependency on H3R2me2a at these target genes. Given that PRMT6 was reported to concomitantly regulate transcriptional initiation and pre-mRNA processing (Harrison et al, 2010; Dowhan et al, 2012), we further compared the PRMT6-dependent splicing and transcriptional response of undifferentiated NT2/D1 cells. Only 25 alternatively spliced genes were also differentially expressed in NT2/D1 KO cells, indicating that the minority of splicing targets (14.5%) showed concomitant transcriptional regulation by PRMT6 (Fig 2B). Among these co-targets of PRMT6-mediated transcriptional initiation and splicing function, only five genes displayed overlapping promoter- and splice site–associated H3R2me2a chromatin marks. These were predominantly mutually exclusive first exons (data not shown), which rely on a combination of alternative transcriptional start sites and AS. Altogether, these results show that most of the PRMT6-dependent splicing targets in NT2/D1 cells are influenced

---

**Figure 1. PRMT6-regulated alternative splicing (AS) in undifferentiated/pluripotent NT2/D1 cells.**
**(A)** Undifferentiated (–, in the absence of ATRA) NT2/D1 control (CT–) or PRMT6 knockout (KO–) cell lines used to perform RNA-seq (Bouchard et al, 2018) and subsequent AS analyses. The scheme was created with BioRender.com. **(B)** Total number of local splicing variations or AS events and the corresponding number of genes differentially spliced in NT2/D1 PRMT6 KO– versus CT– cells using MAJIQ (cutoffs: ΔPSI ≥ 0.05, confidence interval = 0.9). **(B, C)** Pie chart showing the frequency (in percentage and event number) of the 233 PRMT6-regulated AS events (from (B)) with respect to the indicated five major AS categories. **(D, E, F)** Scheme of representative cassette exon events depicted in VOILA (left panels) for *PRMT4* exon 15 (D), *CHASERR* exon 2 (E), and *FLNB* exon 30 (F). **(D, E, F)** Transcripts with skipped/excluded (ΔE) and retained/included (FL, full-length) exons are shown in different colors (red/blue in (D, F), blue/green in (E)). Violin boxplots (*white* dots show the expected PSI values, with the 25 and 75 percentile of the distribution indicated) represent the exon exclusion/inclusion levels in NT2/D1 CT– and PRMT6 KO– cells (–, in the absence of ATRA). The differences [(KO–)–(CT–)] of exclusion/inclusion (ΔPSI/ΔΨ) are shown in the middle. **(D, E, F)** Agarose gel electrophoresis of a representative standard RT–PCR (right panels) showing for *PRMT4* exon 15 (D) and *CHASERR* exon 2 (E) increased exclusion (ΔE) and for *FLNB* exon 30 (F) increased inclusion (FL) in NT2/D1 KO– versus CT– cells. Upper bands correspond to the FL isoform with the retained exon, whereas lower bands represent the ΔE isoform with the exon excluded. Two primer pairs indicated by arrows were designed to specifically amplify the FL and ΔE isoforms. **(E, F)** Corresponding RT–PCR of *GAPDH* (500 bp amplicon) as a housekeeping gene is depicted (of note, this control PCR is identical in (E, F)). Size markers (M) are shown on the left. **(G, H, I)** Quantification of the ΔE isoforms' expression in CT– and KO– cells by independent standard RT–PCR experiments for *PRMT4* ΔE15 (G), *CHASERR* ΔE2 (H), and *FLNB* ΔE30 (I). Transcript levels were plotted in percentage relative to the corresponding FL isoform. Shown are the means ± SD, n = 3–4, *P ≤ 0.05, **P ≤ 0.01, and ****P ≤ 0.0001 using a t test. The data points derived from the same biological replicate are displayed in the same color. **(J)** Expression analysis of PRMT6 in lysates of NT2/D1 KO cells stably expressing doxycycline-inducible WT or enzymatically inactive mutant (MUT) PRMT6 by Western blot. Cells were treated for 3 d with ± doxycycline (Doxy). The antibodies used for immunostaining are indicated on the right, and size markers are shown on the left. **(K, L, M)** Quantification of *PRMT4* ΔE15 (K), *CHASERR* ΔE2 (L), and *FLNB* ΔE30 (M) in NT2/D1 KO– cells re-expressing either WT or enzymatically inactive mutant (MUT) PRMT6 upon doxycycline treatment for 3 d (±Doxy) by independent RT–qPCR experiments. The expression was normalized to full-length (FL) *PRMT4* and depicted relative to the –Doxy conditions, which were set to 1. Shown are the means ± SD, n = 2–3, *P ≤ 0.05 and ns (not significant) using a t test.
Source data are available for this figure.

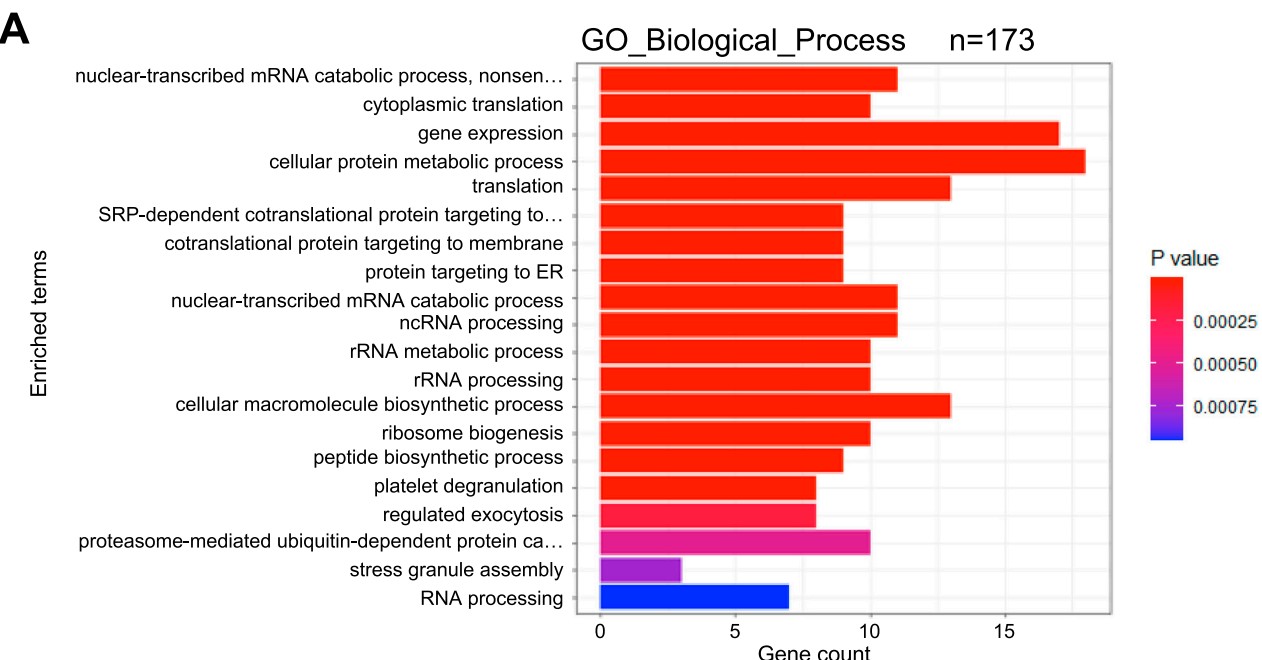

**A**

GO_Biological_Process    n=173

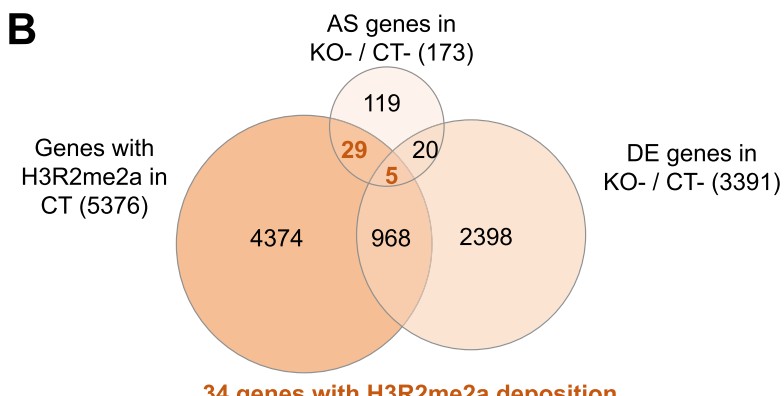

**B**

34 genes with H3R2me2a deposition
proximal to the splice site

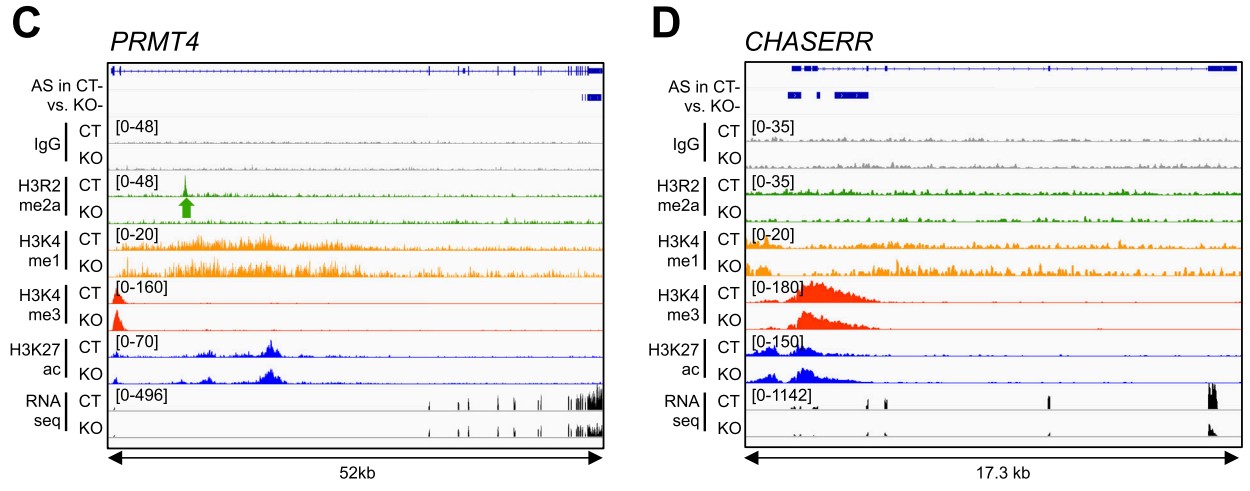

**C** *PRMT4*    **D** *CHASERR*

in an H3R2me2a-independent manner and are not simultaneously transcriptionally regulated by PRMT6. Notably, for a subset of splicing targets, the PRMT6-dependent deposition of H3R2me2a coincides with the AS event, suggesting a potential dual yet distinct function of H3R2me2a in the regulation of transcription and post-transcriptional processes.

### PRMT6-dependent alternative splicing is important for neuronal differentiation of NT2/D1 cells

NT2/D1 cells differentiate into neuronal cells upon treatment with ATRA (Fig 3A). This differentiation is accompanied by the down-regulation of pluripotency genes, such as *OCT4*, *NANOG*, and *SOX2* (Fig S8A), and the up-regulation of neurogenesis-related transcription factor genes, such as the rostral *HOXA* cluster, *RARB*, *MEIS1*, and *MEIS2* (Fig S8B). Given that our previous work revealed that PRMT6 is an important transcriptional co-regulator of the gene expression program in NT2/D1 cells and contributes to their ATRA-induced neuronal differentiation by regulating transcriptional initiation of pluripotency and differentiation genes (Hyllus et al, 2007; Stein et al, 2016; Bouchard et al, 2018), we next investigated the AS response associated with neuronal differentiation of NT2/D1 cells and whether this process would be affected by PRMT6. To achieve this, we first determined the ATRA-regulated LSVs in NT2/D1 cells by comparing the RNA-seq data sets obtained from PRMT6 WT (CT) cells, which were either not treated and pluripotent (CT−) or treated with ATRA and differentiating cells (CT+) (Fig 3A). Thereby, we identified 125 differentiation-associated AS events within 95 genes upon ATRA induction (Fig 3B, Table S1). To define the role of PRMT6 in differentiated NT2/D1 cells, the splicing differences between ATRA-treated CT+ versus KO+ cells were quantified (Fig 3A). This resulted in the identification of 54 AS events within 38 genes (Fig 3B, Table S1). In both cases, the differentiation-associated AS events and the PRMT6-regulated AS events in differentiation, we found that cassette exon alterations prevailed upon PRMT6 deletion (Fig S9A and B), similar to our findings in undifferentiated NT2/D1 cells. Most of these AS targets were not differentially expressed upon ATRA-induced neuronal differentiation (Fig S10A), and their splice sites rarely overlapped with the H3R2me2a mark (Fig S10B), indicating also here a predominant transcription- and H3R2me2a-independent splicing regulation.

In order to elucidate the PRMT6's contribution to differentiation-associated AS, we compared the 125 ATRA-related and the 54 PRMT6-related AS events and found an overlap of 24 events between the two groups (Fig 3C). It is noteworthy that the splicing changes (ΔPSI) induced by ATRA-mediated differentiation (CT− versus CT+) and by PRMT6 knockout in differentiating cells (CT+ versus KO+) exhibited a significant negative correlation, with most of the differentiation-associated splicing effects being reversed in

KO+ cells (Fig 3D). This suggests an involvement of PRMT6 in the differentiation-associated AS of NT2/D1 cells. GO enrichment analysis of the differentiation-associated splicing targets uncovered overrepresented terms such as post-translational modification, histone methylation, mRNA processing, DNA replication, and developmental processes (Fig 3E).

Among the PRMT6-regulated differentiation-associated splicing targets, we identified important histone and chromatin modifiers, for example, ASH2L and DNMT3B, of which the predominant splice isoforms found in ATRA-treated NT2/D1 cells (*ASH2L* E1a and *DNMT3B* ΔE10) have been reported to coincide with a loss of pluripotency and a gain of differentiation (Wang et al, 2001; Gopalakrishna-Pillai & Iverson, 2011; Li et al, 2018). We validated the differentiation-associated splicing effects caused by PRMT6 for *ASH2L*, *DNMT3B* E10 (Fig 4A–D), and several other splicing targets (Fig S11A–F) using standard RT–PCR analysis. Upon ATRA-induced differentiation of CT cells (CT+ versus CT−), the level of the *ASH2L* E1a isoform increased relative to the E1b isoform, whereas KO+ cells revealed a dominance of the E1b isoform similar to the situation in undifferentiated CT− cells (Fig 4A and B). In case of *DNMT3B*, the differentiation-induced preference of exon 10 exclusion (ΔE10) over exon 10 inclusion (FL) was detected in CT cells (CT− versus CT+), but was not observed upon PRMT6 knockout (CT+ versus KO+) (Fig 4C and D). In summary, we confirmed for a number of candidate targets that PRMT6 plays an important role in alternative splicing during neuronal differentiation of NT2/D1 cells.

Finally, to independently examine the contribution of PRMT6 to pluripotency and differentiation decisions, we determined by RT–qPCR the expression levels of the pluripotency markers *OCT4* and *NANOG* in NT2/D1 KO cells without or with the re-expression of PRMT6. As expected (Bouchard et al, 2018), ATRA treatment reduced the expression of *OCT4* and *NANOG* in KO cells, which was further decreased in response to the re-expression of PRMT6 WT (Fig 5A and B). Consistent with this PRMT6-mediated loss of pluripotency of NT2/D1 cells, the re-expression of PRMT6 WT, but not its mutant form, increased the occurrence of the differentiation-associated ΔE10 isoform of *DNMT3B* (Fig 5C). This finding indicates that also during neuronal differentiation, PRMT6 regulates gene expression on the level of AS in a methyltransferase-dependent manner.

To gain some mechanistic insights into how PRMT6 might affect AS, we revisited published interactome studies showing that PRMT6 interacts with numerous RBPs and SFs, such as several HNRNP proteins, NF45/ILF2 and NF90/ILF3 (Avasarala et al, 2020; Schneider et al, 2021; Wei et al, 2021). Interestingly, NF45 has recently been reported to be methylated in a PRMT6-dependent manner and to contribute to AS of cassette exons and mutually exclusive exons, which are the predominant splicing functions of PRMT6 identified here in NT2/D1 cells (Haque et al, 2023; Wu et al, 2024). To address

---

**Figure 2. Potential functional connection between alternative splicing (AS), histone modifications, and transcriptional initiation regulated by PRMT6 in pluripotent NT2/D1 cells.**
**(A)** Top 20 of GO terms found enriched in the 173 PRMT6-regulated alternatively spliced target genes in undifferentiated NT2/D1 (CT−) cells. **(B)** Venn diagram illustrating the intersection of alternatively spliced (AS) and differentially expressed (DE) genes in NT2/D1 KO− versus CT− cells and H3R2me2a deposition proximal to the splice site in CT cells. **(C, D)** Genome browser views of AS events, ChIP-seq data sets of IgG control (gray), H3R2me2a (green), H3K4me1 (yellow), H3K4me3 (red), H3K27ac (blue), and RNA-seq tracks (black) (Bouchard et al, 2018) for the *PRMT4* (C) and *CHASERR* (D) gene locus in NT2/D1 CT and KO cells (−, in the absence of ATRA). **(C)** H3R2me2a enrichment peak in the *PRMT4* gene locus is highlighted by a green arrow below the H3R2me2a/CT track in (C). ChIP scales are indicated in brackets in the upper left corner of the tracks. The distance (in kb) of the considered region is depicted below the browser views.

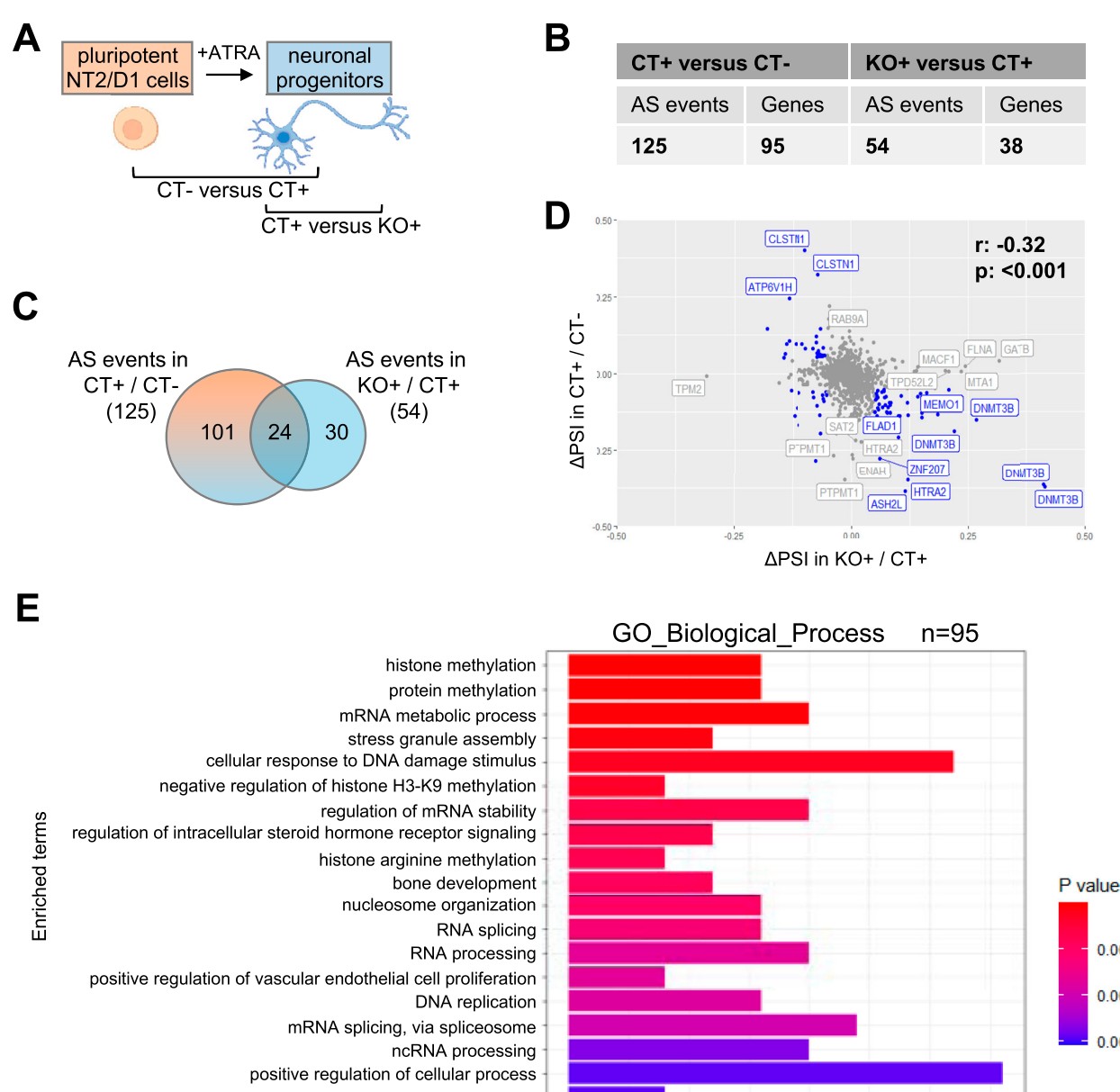

**Figure 3. PRMT6-dependent alternative splicing (AS) upon ATRA-induced neuronal differentiation of NT2/D1 cells.**
**(A)** Pluripotent (−, in the absence of ATRA) NT2/D1 control cells (CT−) and differentiating (upon ATRA treatment) control cells (CT+), as well as differentiating PRMT6 knockout cells (KO+) analyzed for AS during differentiation (CT− versus CT+) and the splicing contribution of PRMT6 in differentiation (CT+ versus KO+) using RNA-seq. The scheme was created with BioRender.com. **(B)** Total number of local splicing variations or AS events and the corresponding number of genes differentially spliced in NT2/D1 CT+ versus CT− and in KO+ versus CT+ cells using MAJIQ (cutoffs: ΔPSI ≥ 0.05, confidence interval = 0.9). **(C)** Venn diagram illustrating the intersection of differentiation-associated AS events (CT+/CT−) and PRMT6-regulated splicing events in differentiation (KO+/CT+). **(D)** Scatter plot displaying the ΔPSI values of AS events identified in CT+/CT− versus AS events in KO+/CT+ (cutoffs: in blue, AS events with ΔPSI value ≥ 0.05, and in gray, AS events with ΔPSI value < 0.05, confidence value = 0.9). The Spearman correlation coefficient r and the *P*-value are indicated. **(E)** Top 20 of GO terms found enriched in the 95 alternatively spliced genes upon differentiation of CT cells.

whether PRMT6 and NF45 might associate with each other also in NT2/D1 cells, we carried out co-immunoprecipitation assays and found that the two proteins interact at the endogenous level (Fig 5D). We could not detect an interaction between PRMT6 and NF90, the heterodimerization partner of NF45 (data not shown). To

determine whether NF45 also serves as a substrate for PRMT6 in NT2/D1 cells, we performed immunoprecipitation of NF45 from NT2/D1 CT and KO cell lysates followed by Western blot analysis for asymmetric dimethylarginine (ADMA) (Fig 5E). Our results showed that the precipitated/purified NF45 stains for ADMA in CT lysates

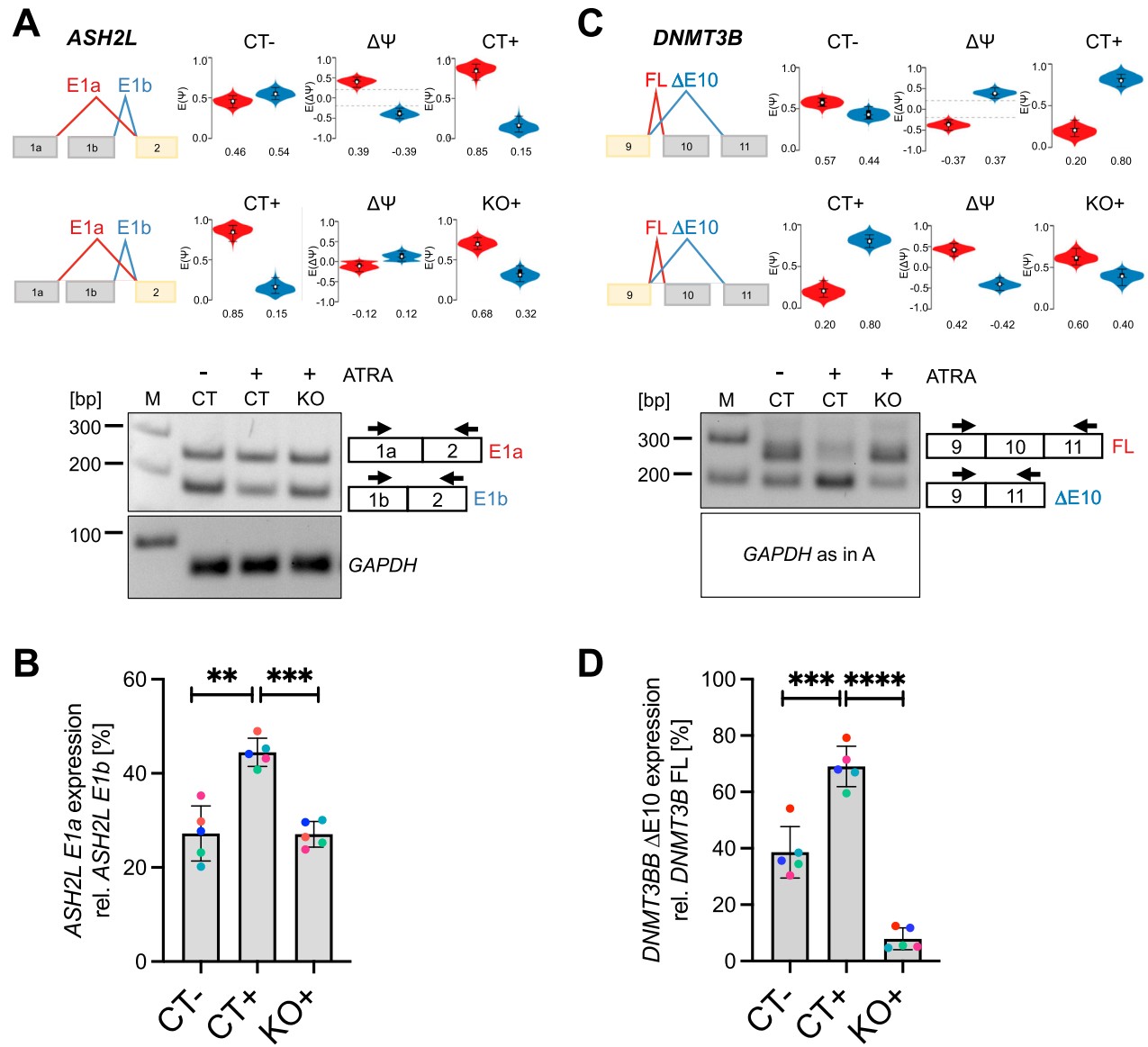

**Figure 4. PRMT6-dependent alternative splicing (AS) of *ASH2L* and *DNMT3B* upon ATRA-induced neuronal differentiation of NT2/D1 cells.**
**(A, C)** Scheme of AS events depicted in VOILA (upper panels) for *ASH2L* mutually exclusive exon 1 (E1a/b) (A) and *DNMT3B* exon 10 (C). Splice isoforms are shown in different colors (red/blue). Violin boxplots (*white* dots show the expected PSI values, with the 25 and 75 percentile of the distribution indicated) represent the alternative exon levels in NT2/D1 CT− versus CT+ and CT+ versus KO+ cells. The splicing differences (ΔPSI/ΔΨ) are each shown in the middle. **(A, C)** Agarose gel electrophoresis of a representative standard RT–PCR (lower panels) showing alterations for *ASH2L* E1a/1b (A) and for *DNMT3B* E10 (C) in NT2/D1 CT−, CT+, and KO+ cells. Upper bands correspond to the *ASH2L* E1a isoform or the *DNMT3B* full-length (FL, with retained exon) isoform, whereas lower bands represent the *ASH2L* E1b isoform or the exon 10–excluded (ΔE10) *DNMT3B* isoform. Two primer pairs indicated by arrows were designed to amplify the specific isoforms. **(A, C)** Corresponding RT–PCR of *GAPDH* (70 bp amplicon) as a housekeeping gene is depicted (of note, this control PCR is identical in (A, C)). Size markers (M) are shown on the left. **(B, D)** Quantification of the alternatively spliced isoforms in CT−, CT+, and KO+ cells by independent standard RT–PCR experiments for *ASH2L* E1a (B) and *DNMT3B* ΔE10 (D). Transcript levels were plotted in percentage relative to the corresponding alternative splice isoform. Shown are the means ± SD, n = 5, **$P ≤ 0.01$, ***$P ≤ 0.001$, and ****$P ≤ 0.0001$ using a *t* test. The data points derived from the same biological replicate are displayed in the same color.
Source data are available for this figure.

and this staining is diminished in the KO condition, indicating that NF45 is methylated in a PRMT6-dependent manner in NT2/D1 cells. These findings suggest that PRMT6 might potentially regulate AS of cassette exons and mutually exclusive exons through the interaction and modification of splicing modulators, such as NF45, in NT2/D1 cells.

In summary, our data identify PRMT6 as a regulator of neurogenesis that influences the gene expression program of neural pluripotent progenitors and differentiating neurons, as exemplified for NT2/D1 cells, in at least two ways: firstly, by fine-tuning the process of transcriptional initiation (Hyllus et al, 2007; Bouchard et al, 2018); and secondly, by modulating mRNA splicing.

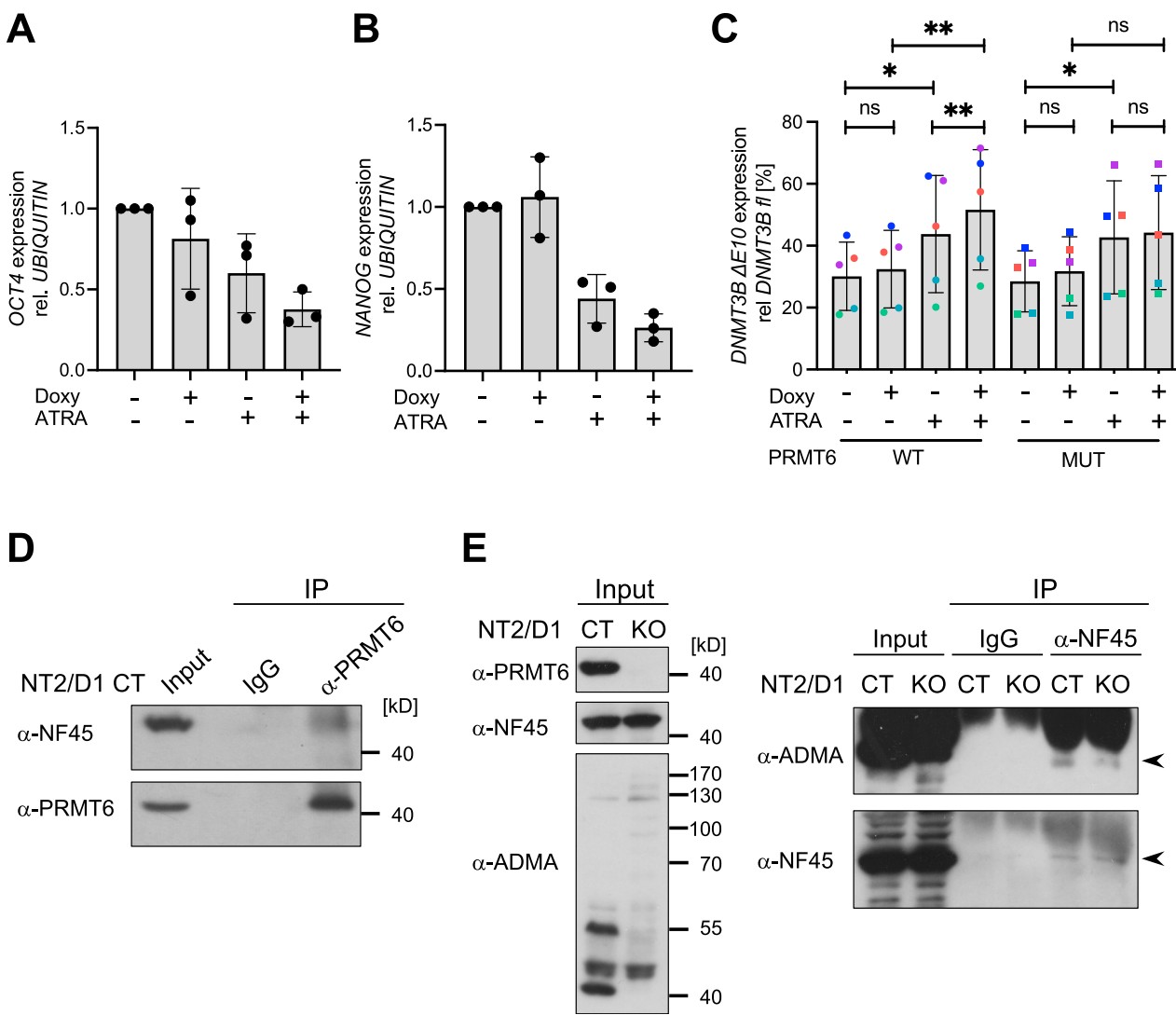

**Figure 5. Influence of PRMT6's catalytic activity on the *DNMT3B* splice isoform switching in differentiating NT2/D1 cells.**
**(A, B)** Quantification of the transcript levels of the pluripotency markers *OCT4* (A) and *NANOG* (B) in NT2/D1 PRMT6 knockout (KO) cells re-expressing WT PRMT6 upon doxycycline treatment (±Doxy) and subsequent induction of differentiation (±ATRA) by independent RT–qPCR experiments. *OCT4* and *NANOG* were normalized to *UBIQUITIN* and depicted relative to the −Doxy/−ATRA condition, which was set to 1. Shown are the means ± SD, n = 3. **(C)** Quantification of *DNMT3B* ΔE10 transcript levels in NT2/D1 KO cells re-expressing either WT or enzymatically inactive mutant (MUT) PRMT6 upon doxycycline treatment (±Doxy) and subsequent induction of differentiation (±ATRA) by independent RT–qPCR experiments. *DNMT3B* ΔE10 transcript levels were plotted in percentage relative to the corresponding full-length (FL) isoform. Shown are the means ± SD, n = 5, *$P \leq 0.05$, **$P \leq 0.01$, and ns (not significant) using a *t* test. The data points derived from the same biological replicate are displayed in the same color. **(D)** PRMT6-NF45 interaction analysis in NT2/D1 CT cell lysates using α-PRMT6 antibodies or IgG as a negative control in immunoprecipitation (IP) assays. IP reactions and input lysates were separated by SDS–PAGE and analyzed by immunoblotting using the indicated antibodies on the left. Size markers are shown on the right. **(E)** Analysis of PRMT6-dependent arginine methylation of NF45 in NT2/D1 CT and KO cell lysates using α-NF45 antibodies or IgG as a negative control in immunoprecipitation (IP) assays. Input lysates (left panel) and IP reactions (including as well the input samples, right panel) were separated by SDS–PAGE and analyzed by immunoblotting using the indicated antibodies on the left. Arrowheads (right panel) mark the NF45 protein band. Size markers are shown in the left panel. Source data are available for this figure.

# Discussion

PRMTs are well characterized for their epigenetic function, which is attributed to their ability to methylate histones and other chromatin-associated proteins (Rakow et al, 2020). Thus, arginine methylation influences chromatin structure and can lead to both transcriptional activation and repression. A second very prominent nuclear function of PRMTs was discovered through profiling of the cellular arginine methylome, which revealed substantial arginine

methylation of RBPs, such as SFs, specifically in their RG/RGG-repeat domains (Larsen et al, 2016). Arginine methylation of RBPs exerts a significant influence on their capacity for protein–protein interaction and RNA-binding ability, which is of critical importance for mRNA splicing, RNA localization/stability and translation (Guccione & Richard, 2019; Wei et al, 2021). A key role in regulating pre-mRNA processing has been established for PRMT5 through arginine methylation of Sm proteins, thereby ensuring spliceosomal assembly (Meister & Fischer, 2002; Bezzi et al, 2013). Type I PRMT

family members, in particular PRMT1 and PRMT4, are as well implicated in pre-mRNA maturation through modification of selected SFs (Côté et al, 2003; Cheng et al, 2007).

In addition to directly regulating the cellular splicing machinery by methylating RNA processing components, PRMTs might also coordinate transcription and downstream splicing events through their histone-modifying activity at target gene loci. In agreement with this notion, splicing is known to be a co-transcriptional process affected by chromatin and transcriptional co-regulators. Histone modification patterns enriched at splice sites have been found to enable chromatin recruitment of splicing factors, thus determining splicing decisions (Luco et al, 2010). Remarkably, SACS encompass, among others, the histone marks H3K4me1, H3K4me3, and H3K27ac, whose deposition was found to be influenced by PRMT6 and its major histone substrate H3R2 (Hyllus et al, 2007; Sims et al, 2007; Bouchard et al, 2018; Agirre et al, 2021; Segelle et al, 2022).

Therefore, we investigated here in a cell model of neuronal differentiation, in which promoter- and enhancer-associated dimethylation of H3R2 by PRMT6 regulates transcriptional initiation and establishes a differentiation-relevant gene expression program (Bouchard et al, 2018), whether PRMT6 impacts RNA processing on the level of AS, potentially in an H3R2me2a-dependent manner. Our results demonstrate that PRMT6 deletion results in widespread alterations of AS in pluripotent and differentiating NT2/D1 cells, with the greatest impact observed on cassette exons and intron retentions. Similar to our previous findings on the methyltransferase dependence of PRMT6's transcriptional function, the enzyme requires its enzymatic activity also for splicing regulation. However, only a minority of the PRMT6-dependent transcriptional and splicing targets overlap, suggesting no generally simultaneous control of both processes by PRMT6 in NT2/D1 cells, in contrast to results received in breast cancer cells (Dowhan et al, 2012). On the mechanistic level, we found that PRMT6 might influence AS of a small subset of target genes directly through H3R2me2a deposition, as the corresponding splice sites revealed an enrichment of H3R2me2a at the chromatin level. Other marks adjacent to H3R2me2a, such as H3K4me1, H3K4me3, and H3K27ac, did not coincide with these splicing events, making this type of H3R2me2a-containing SACS unlikely. Most of the PRMT6-regulated splicing events in NT2/D1 cells did not show overlapping H3R2me2a deposition. These H3R2me2a-independent splicing functions of PRMT6 might be executed by methylation events in other PRMT6 substrates, such as different methylation sites in histones (H3R42, H2AR29) or methylation of splicing factors, as well as splicing modulators, as our results suggest for NF45 (Waldmann et al, 2011; Casadio et al, 2013; Wei et al, 2021).

Among the PRMT6-dependent splicing targets, we identified the transcripts of a number of important histone and chromatin modifiers. These include *PRMT4*, *ASH2L*, and *DNMT3B*, whose gene products are impacted by the PRMT6-regulated splicing effect in a functional manner. With respect to the *PRMT4* splicing event, PRMT6 promotes E15 inclusion and thus the generation of the *PRMT4* FL transcript, which is the major transcript isoform in the brain tissue and encodes the automethylation site R551 (Kuhn et al, 2011; Wang et al, 2013). Thereby, PRMT6 seems to foster the functionality of the PRMT4 protein in pluripotent cells, in

agreement with PRMT4's role in pluripotency-related gene expression (Wu et al, 2009). In ATRA-treated differentiating NT2/D1 cells, PRMT6 regulates splicing events, such as the mutually exclusive E1a choice in *ASH2L* and E10 exclusion in *DNMT3B*, both of which have been reported to coincide with a loss of pluripotency and a gain of differentiation (Wang et al, 2001; Gopalakrishna-Pillai & Iverson, 2011; Li et al, 2018). These findings on splicing contributions are consistent with the transcriptional output of PRMT6, as the enzyme also regulates key steps in neuronal differentiation at the level of transcriptional activation and repression (Bouchard et al, 2018).

# Materials and Methods

### Cell lines and reagents

All the cells were maintained in DMEM supplemented with 10% FCS (Gibco/BRL) and 1% penicillin/streptomycin at 37°C and 5% $CO_2$. NT2/D1, HeLa, HEK293T, and U2OS CT cells, which contain WT *PRMT6* similar to the parental cells (CRISPR/Cas9 control/CT, stably infected with GFP gRNAs; Bouchard et al, 2018), and the corresponding KO cells, in which *PRMT6* is mutated and the gene function is deleted (CRISPR/Cas9-mediated knockout/KO of PRMT6, stably infected with PRMT6 gRNAs; Bouchard et al, 2018), were in addition cultured in the presence of 1 μg/ml puromycin (InvivoGen). Neuronal differentiation of NT2/D1 cells was induced with 1 μM ATRA (Sigma-Aldrich) according to the protocol of Andrews (1984). For the doxycycline-inducible re-expression of WT PRMT6 or enzymatically inactive mutant (MUT) PRMT6 in NT2/D1 KO cells, correspondingly infected rescue cell pools were continuously cultured in the presence of 700 μg/ml G418 (InvivoGen) and treated for 3 d with 1 μg/ml doxycycline (D9891; Sigma-Aldrich). The following antibodies were used: rabbit anti-PRMT6 (Bouchard et al, 2018), mouse anti-PRMT6 (sc-271744; Santa Cruz), rabbit anti-ADMA (13522; Cell Signaling), rabbit anti-NF45 (NJ161, gift from Mike Mathews) (Parrott et al, 2005), mouse anti-VINCULIN (sc-73614; Santa Cruz), and mouse anti-β-ACTIN (A5441; Sigma-Aldrich).

### Production of lentiviral particles and infection of cells

Human full-length WT and enzymatically inactive mutant (MUT, VLD/KLA; Neault et al, 2012) PRMT6 cDNAs with C-terminal HA- and V5-tag (HA-PRMT6) were amplified by PCR, cloned into the NcoI and XhoI sites of the pENTR4 entry vector, and then transferred into pInducer20 via LR recombination (Gateway Cloning System, Invitrogen). For the re-expression of HA-PRMT6 in NT2/D1 KO cells, HEK293T cells were transfected with the two packaging plasmids pMD2.G and psPAX2 together with the lentiviral expression plasmid pInducer20 encoding either HA-PRMT6 WT or MUT. Transfections were performed using Fugene (Roche). Supernatants containing lentiviral particles were harvested 1–2 d after transfection. NT2/D1 KO cells were infected in the presence of polybrene (8 μg/ml) with viruses containing either HA-PRMT6 WT or MUT. Rescue cell pools were selected using 700 μg/ml G418 (InvivoGen).

## Protein isolation, immunoprecipitation, and Western blot

For lysis, cells were resuspended in IPH buffer (50 mM Tris–HCl, pH 8.0, 200 mM NaCl, 5 mM EDTA, 0.5% [vol/vol] NP-40) plus protease inhibitors (10 µg/ml of aprotinin, leupeptin, and PMSF) followed by three times freezing and thawing. Cell extracts were subjected to Benzonase treatment (0.25 U/µl in the presence of 7.5 mM $MgCl_2$ for 30 min at 4°C). After centrifugation, cell lysates were either directly analyzed by SDS–PAGE followed by immunoblotting or subjected to immunoprecipitation. In the latter case, 1–1.5 mg of protein was incubated with 8–12 µg of the indicated antibodies overnight at 4°C. Then precipitates were bound to BSA-blocked Protein A-Sepharose (GE Healthcare), washed four times with IPH buffer, and finally analyzed together with 1–5% input lysate by SDS–PAGE followed by immunoblotting.

## RNA isolation, RT–PCR, and reverse transcription–quantitative PCR (RT–qPCR)

Total RNA was isolated using GenUP Total RNA Kit (biotechrabbit). For standard RT–PCR, cDNA synthesis was carried out with 0.5–1 µg RNA as a template using qScript cDNA Synthesis Kit (Quantabio), which contains a mixture of random hexamer and oligo-dT primers, according to the manufacturer's instruction. Standard PCR was performed using Phusion High-Fidelity DNA Polymerase (NEB) according to the manufacturer's protocol. Products of the standard PCR were subjected to agarose gel electrophoresis (2.5%) and visualized with the Intas gel documentation system. Quantification of the isoforms was conducted with ImageJ/Fiji (NIH).

For RT–qPCR, cDNA synthesis was performed using 0.5–1 µg of total RNA as a template together with oligo-(dT) primers and M-MLV reverse transcriptase (Thermo Fisher Scientific) according to the manufacturer's instruction. cDNA was then subjected to quantitative amplification using Absolute qPCR SYBR Green Mix (Thermo Fisher Scientific) and CFX Connect Real-Time System (Bio-Rad). The quantitative analysis of the splice variants was performed as described by Harvey & Cheng (2016).

The following listed forward (fwd) and reverse (rev) primers were used for RT–PCR and/or RT–qPCR of the indicated human gene transcripts.

| Name | Sequence (5'–3') | PCR type |
| --- | --- | --- |
| ASH2L E1a fwd | GAGAAGTCCAGGGGTGGC | PCR |
| ASH2L E1b fwd | TTAACGGGAGGCCTTCACAT | PCR |
| ASH2L E2 rev | CACCGCTTACATCGACCAAG | PCR |
| CALD1 E3a fwd | AAAGAGGGAGGAGATGCGAC | PCR |
| CALD1 E3b fwd | GTCTCCGTATCTCTCTGCCC | PCR |
| CALD1 E4 rev | CTGTTCTGGGCATTCACCTC | PCR |
| CHASERR E1 fwd | GACTCGGGTTTGGGCGAC | PCR |
| CHASERR E4 rev | GTTCAGAACCAACAGCAACCA | PCR |
| CHASERR E2 fwd | ACCTCTTTCCGCCGATTTTG | qPCR |
| CHASERR E2 rev | TCACCTTCGCATGTATCACTG | qPCR |
| CHASERR ΔE2 fwd | GGACCCGCCGATTTTGAAAC | qPCR |

**Continued**

| Name | Sequence (5'–3') | PCR type |
| --- | --- | --- |
| CHASERR ΔE2 rev | TCAGAACCAACAGCAACCAA | qPCR |
| DNMT3B E10 fwd | TCGTGCAGGCAGTAGGAAAT | qPCR |
| DNMT3B E10 rev | TCGGCTCTGATCTTCATCCC | qPCR |
| DNMT3B ΔE10 fwd | CGCAACCAGAGAACAAGACTCG | qPCR |
| DNMT3B ΔE10 rev | CGGCTCTGATCTTCATCCCC | qPCR |
| DNMT3B E9 fwd | CAGCCCTGGAGACTCATTGG | qPCR |
| DNMT3B E9 rev | GTTGCGTGTTGTTGGGTTTG | qPCR |
| DNMT3B E9 fwd | CAGCCCTGGAGACTCATTGG | PCR |
| DNMT3B E11 rev | ATTTGTCTTGAGGCGCTTGG | PCR |
| DNMT3B E20 fwd | GCCATCAAAGTTTCTGCTGC | PCR |
| DNMT3B E23 rev | GACCTTCCCAGCAGCTTCT | PCR |
| FLNB E29 fwd | GATCTATGTGCGCTTCGGTG | PCR |
| FLNB E31 rev | GCCGTTCATGTCACTCACTG | PCR |
| FLNB E30 fwd | ATGCATGTCCCCCTGGATTC | qPCR |
| FLNB E30 rev | CAGGCCGTTCATGTCACTCA | qPCR |
| FLNB ΔE30 fwd | TGTCATGGTGACCGAAGAGG | qPCR |
| FLNB ΔE30 rev | CAGCAAACGGAATGACCAGG | qPCR |
| GAPDH (500 bp) fwd | ACCACAGTCCATGCCATCAC | PCR |
| GAPDH (500 bp) rev | TCCACCACCCTGTTGCTGTA | PCR |
| GAPDH (70 bp) fwd | AGCCACATCGCTCAGACAC | PCR |
| GAPDH (70 bp) rev | GCCCAATACGACCAAATCC | PCR |
| HTRA2 E6 fwd | GGTGATGATGCTGACCCTGA | PCR |
| HTRA2 E8 rev | CCCCAATGGCCAAAATCACA | PCR |
| NANOG fwd | TCCAGCAGATGCAAGAACTC | qPCR |
| NANOG rev | TTGCTATTCTTCGGCCAGTT | qPCR |
| OCT4 fwd | GACAACAATGAAAATCTTCAGGAGA | qPCR |
| OCT4 rev | TTCTGGCGCCGGTTACAGAACCA | qPCR |
| PRMT4 E13 fwd | CAGCACCTACAACCTCAGCA | PCR |
| PRMT4 E16 rev | GGCTGTTGACTGCATAGTGG | PCR, qPCR |
| PRMT4 E15 fwd | ATGAGCACGGGGATTGTCCAA | qPCR |
| PRMT4 ΔE15 fwd | CCTGATTCCTTTAGGGTCCTCC | qPCR |
| PRMT4 ΔE15 rev | CTAGCTCCCGTAGTGCATGG | qPCR |
| PRMT6 endo fwd | AGGGGAGTCGGAGAAACC | qPCR |
| PRMT6 endo rev | CGCCTGTTTCCAGTGAGTG | qPCR |
| PRMT6 exo fwd | CCGGTGTAGGAGGACGATTT | qPCR |
| PRMT6 exo rev | TTGGACAGGCTCGTTCAGAT | qPCR |
| TFDP1 E10 fwd | TGCTCTGCCGAAGACCTTAA | PCR |
| TFDP1 E12 rev | CCGCTGTACTGAGACCCATT | PCR |
| ZFAS1 E2 fwd | CGTGCAGACATCTACAACCT | PCR |
| ZFAS1 E5 rev | AGGGCTCCTCTCATATTCCA | PCR |
| ZNF207 E7 fwd | CCCACAGCCTCCAGTTACTA | PCR |
| ZNF207 E9 rev | GCAGGGAATGTAGGCTTTGG | PCR |
| UBI fwd | CACTTGGTCCTGCGCTTGA | qPCR |
| UBI rev | CAATTGGGAATGCAACAACTTTAT | qPCR |

### RNA-seq and splicing analyses

The generation of the RNA-sequencing (RNA-seq) data was reported in Bouchard et al (2018). Reads of the biological triplicate samples were aligned against the *Homo sapiens* reference genome (version GRCh38) and GENCODE gene annotation (version 36) using STAR (version 2.7.7a) (Dobin et al, 2013) and then parsed through htseq-count (version 0.13.5) (Dobin et al, 2013; Anders et al, 2014), calculating how many RNA-seq reads map to which gene. Differentially regulated genes were calculated using DESeq2 (version 1.26.0) (Love et al, 2014). For differentially regulated genes, a threshold of an absolute fold change ($|FC|$) ≥ 2 and an adjusted *P*-value ≤ 0.01 (Benjamini–Hochberg correction) was followed. RNA-seq read coverage tracks were generated using bamCoverage from deepTools (version 3.5.0) with reads per million mapped reads (RPKM) normalization. The tool Modeling Alternative Junction Inclusion Quantification (MAJIQ, version 2.2) was used for the AS analysis (Vaquero-Garcia et al, 2016). MAJIQ creates LSVs and quantifies AS in each LSV by calculating a so-called PSI value (percent selected index, Ψ) for each exon–exon junction therein. The difference in PSI values between two conditions, called ΔPSI, is the measure of the AS change between two samples. The visualization of the data showing AS changes created by MAJIQ was performed with VOILA (version 2.0.0) (Vaquero-Garcia et al, 2016). Integrative Genomics Viewer (IGV, version 2.10.2) was used to explore and visualize genomic data (Robinson et al, 2011). To analyze the biological significance of splicing targets, enrichR was used for gene ontology analysis (Kuleshov et al, 2016).

### Statistical analysis

All experiments were independently performed at least three times (biological replicates). Error bars represent ± SD of the mean for these replicates. Corresponding statistical tests are mentioned in the figure legends. In case of agarose gels and immunostainings, reproducible and representative data sets are depicted.

## Data Availability

RNA-seq (NT2/D1 CT and KO cells) and ChIP-seq data sets are available at the GEO accession number GSE107612 (Bouchard et al, 2018).

## Supplementary Information

## Acknowledgements

We thank all members of the Bauer laboratory for support and helpful discussion, in particular Silke Caspari and Inge Sprenger for technical assistance. We acknowledge Mario Keller and all members of the Zarnack laboratory for advice and discussion. This work was funded by DFG (Deutsche Forschungsgemeinschaft) grants TRR81 A03, SFB1213 A05, BA 2292/1, BA 2292/4, and BA 2292/5 to U-M Bauer and by the Deutsche José Carreras Leukämie-Stiftung e.V. grant DJCLS 05 R/2020 to U-M Bauer.

## Author Contributions

M Eudenbach: data curation, validation, investigation, visualization, methodology, and writing—review and editing.
J Busam: data curation, software, formal analysis, validation, investigation, visualization, and writing—review and editing.
C Bouchard: conceptualization, data curation, validation, investigation, visualization, methodology, and writing—review and editing.
O Rossbach: conceptualization, resources, methodology, and writing—review and editing.
K Zarnack: conceptualization, data curation, software, supervision, investigation, and writing—review and editing.
U-M Bauer: conceptualization, data curation, formal analysis, supervision, funding acquisition, visualization, project administration, and writing—original draft.

## Conflict of Interest Statement

The authors declare that they have no conflict of interest.

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
