## [Reviewer comments · Life Science Alliance]

Life Science Alliance

Assessment of PRMT6-dependent alternative splicing in pluripotent and differentiating NT2/D1 cells

Matthias Eudenbach, Jonas Busam, Caroline Bouchard, Oliver Rossbach, Kathi Zarnack, and Uta-Maria Bauer
DOI: <https://doi.org/10.26508/lsa.202402946>

Corresponding author(s): Uta-Maria Bauer, Philipps University of Marburg

Review Timeline:	Submission Date:	2024-07-17
	Editorial Decision:	2024-08-26
	Revision Received:	2024-12-23
	Editorial Decision:	2025-01-13
	Revision Received:	2025-01-20
	Accepted:	2025-01-21

Transaction Report:

August 26, 2024

Re: Life Science Alliance manuscript #LSA-2024-02946-T

Prof. Uta-Maria Bauer
Philipps-University Marburg
Institute for Molecular Biology and Tumor Research (IMT)
Emil-Mannkopff Str.2
Marburg 35032
Germany

Dear Dr. Bauer,

Thank you for submitting your manuscript entitled "PRMT6-dependent splicing functions contribute to pluripotency and neuronal differentiation" to Life Science Alliance. The manuscript was assessed by expert reviewers, whose comments are appended to this letter. We invite you to submit a revised manuscript addressing the Reviewer comments.

Thank you for this interesting contribution to Life Science Alliance. We are looking forward to receiving your revised manuscript.

Sincerely,

B. MANUSCRIPT ORGANIZATION AND FORMATTING:

Reviewer #1 (Comments to the Authors (Required)):

The manuscript by Eudenbach et al examines the role of PRMT6 in alternative splicing. PRMT6 is known for its role as a protein arginine methyltransferase that specifically catalyzes the dimethylation of histone H3 at arginine 2 (H3R2me2). This modification is associated with transcriptional repression and plays a crucial role in regulating gene expression. PRMT6 has been implicated in various cellular processes, including DNA repair, cell cycle regulation, and stem cell maintenance. A few studies showed PRMT6 is involved in alternative splicing in cancer cells. However, it is not investigated in differentiation. While the involvement of PRMT6 in regulating gene expression is shown by the author's lab and others before (Bouchard et al 2018), its role in alternative splicing is not explored in the same context. In this manuscript, using their previously published RNA-seq data from NT2/D1 cell lines, the authors show the role of PRMT6 in regulating alternative splicing. The manuscript is well-written and presented and the question asked is relevant to the field. However, there are several concerns that need to be addressed before possible consideration of publication.

Major comments:

1. The complete list of splicing events related to Figure 1 and Figure 3 with details of splicing type and genomic coordinates should be provided as supplementary table.
2. For all semi-quantitative RT-PCRs, a loading control (e.g. ACTB or GAPDH) should be included and normalized when quantifying the amplicons. Sometimes, the band differences may arise from un-uniform loading of samples. For example, in Fig. 1D, the bands in KO sample seems brighter compared to CT.
3. The authors don't mention why they chose to validate PRMT4 and CHASERR. Are these the most significantly spliced?
4. While the PRMT4 splicing levels in Fig. 1D as shown violin plots, more-or-less match with gel images, the CHASERR levels in E doesn't seem significantly different between CT and KO, at least visually. Could authors clarify this?
5. It would be great if authors show RT-PCR validation (gel images) of PRMT4 Ex4 splicing in their rescue experiments (related to Fig1F and G). It will also add strength to the rescue experiments if they include CHASERR Ex2 splicing rescue.
6. Authors mention "H3R2me2 deposition not overlapping with the splice site (e.g. PRMT4, Figure 2C) or even no H3R2me2a enrichment within in the target gene locus (e.g. CHASERR, Figure 2D)". It is very difficult to see whether the deposition overlaps or not at the splice site. Not sure what is the best way to show this, may be zooming in with high-magnification image of the splice site to compare? Is there a way to quantify the deposition for a given locus? If so, it would be very useful to include it.
7. In the scatter plot presented in Fig. 3D, there are genes (e.g. CLSTN1, DNMT3B) are labelled more than once. Are these different splicing events within the same gene? If so, it would be useful if they are included in the validation alongside F and G.
8. Statistics in Fig. 4C is not confusing. In PRMT6 WT, the difference (Dox- and +; ATRA+ +), doesn't look significant at all, but it shown as **, could authors check this and other comparisons as well?
9. PRMT6's splicing-related mechanism is thought to involve methylating RNA binding proteins. So, it is reasonable to say that it would influence a set of RBP methylation to regulate splicing. Authors could analyze some putative RBPs in KO background to probe PRMT6's involvement. Without this the schematic presented in Fig. 4D is too vague.

Minor comments:

1. The authors have performed RNA-seq analysis to reveal alternative splicing using their previously published datasets (Bouchard et al 2018). However, they don't mention in both 2018 and current paper about the read depth (how many million reads). This information is crucial as the read depth influences alternative splicing analysis outcome. It would be great if authors provide this information in the methods.
2. Full uncropped raw images of agarose gel and western blot images should be provided as supplementary images.
3. The title is very generic and might mislead. It could mean pluripotency in ES cells in mouse, human etc. As all the experiments in this paper were done on NT2/D1 cell line (which is not bad), the authors could consider including it in the title and revise it to precisely reflect the work.

Reviewer #2 (Comments to the Authors (Required)):

In this study, Prof. Bauer and colleagues investigate the role of PRMT6, a protein arginine methyltransferase, in modulating

alternative splicing choices. The authors demonstrate that PRMT6, in addition to regulating H3R2 methylation and gene expression, it also influences alternative splicing patterns in the NT2/D1 cancer cell line. Through a combination of mechanistic work and analysis of available CHIP-Seq and RNA-Seq datasets, they show that PRMT6's methyltransferase activity is necessary for its impact on alternative splicing. Interestingly, this effect appears to occur largely independently of histone modification and gene expression regulation. Furthermore, the study reveals that PRMT6 can modulate certain splicing events during the neuronal differentiation of NT2/D1 cells upon retinoic acid treatment. Approximately 20% of the RA-dependent splicing events are also affected by PRMT6 knockout. However, it remains uncertain whether these events are directly regulated by PRMT6's methyltransferase activity or if they are by-products of less efficient neuronal differentiation following PRMT6 knockout.

Overall, the manuscript is well-written and establishes a compelling link between PRMT6 and alternative splicing regulation. However, the observed effect of PRMT6 on splicing regulation appears modest, with as few as 55 splicing events detected under some conditions. Therefore, additional validation of the splicing changes would enhance confidence in the results. Moreover, the manuscript provides limited insights into the mechanisms by which PRMT6 influences splicing. Specific comments and suggestions for improvement are provided below.

1. To strengthen the data, the authors should provide additional validation using RT-PCR and agarose gel assays beyond the two events shown in Figures 1 and 3. Ideally, profiling 10-20 splicing events, providing the corresponding agarose gels in the supplementary materials, and generating correlation scatter plots between RNA-Seq-derived and RT-PCR-derived PSI values would substantially bolster the validity of the findings.
2. The study offers limited mechanistic insights into how PRMT6 affects alternative splicing. While identifying a specific PRMT6-targeted splicing factor might be beyond the scope of this study, the authors should at least attempt to determine whether the observed splicing changes are a direct result of PRMT6's methyltransferase activity or if they are secondary to altered differentiation status in PRMT6 knockout cells.
3. The manuscript does not adequately address how PRMT6 affects NT2/D1 cell differentiation. To clarify this, the authors should expand Figure 4A-B to include wild-type (WT) cells, not just the knockout (KO) cells. Additionally, monitoring a neuronal differentiation marker, such as TUBB3, would provide further clarity on PRMT6's role in differentiation.

Minor comments:

4. Gene expression tracks should be included in Figures 4C-D and supplementary Figures EV4A-C to provide a more comprehensive view of the data.
5. The ribosomal proteins highlighted in Extended Data Figure EV4 could be among the genes validated by RT-PCR assays, as suggested in the major comment 1.
6. A representative agarose gel corresponding to Figure 4C could be included either as a supplementary figure or positioned above the bar plot for clearer visualization.

Reviewer #3 (Comments to the Authors (Required)):

I am happy to review the manuscript "PRMT6-dependent splicing functions contribute to pluripotency and neuronal differentiation" by Eudenbach et al. In their project the authors examine the function of the protein arginine methyltransferase 6 (PRMT6), an enzyme known to methylate arginine residues in histone and non-histone proteins. The focus is on the involvement of PRMT6 in alternative splicing during neuronal development. They performed loss of function experiments to identify PRMT6 dependent spliced genes. They conclude that PRMT6 influences splicing of neuronal genes in a histone methylation dependent and independent manner, which contributes to neuronal differentiation and pluripotency.

The manuscript is well written and the figures are mostly clear. The conclusions are mostly well supported by the data and interesting in the context of neuronal differentiation. However, the observation that PRMT6 is involved in splicing control is not novel and mechanistic data are elusive.

Main Points

- The mechanisms how PRMT6 influences splicing is elusive. This limits the impact of the data.
- It is not clear if differential PRMT6 dependent splicing contributes to neuronal differentiation. Does the alternatively spliced DNMT contribute?
- The title mentions an effect of alternative PRMT6 dependent splicing on pluripotency. Do the data really support this notion?
- Mostly available data are reanalyzed. Which is fine. However, are the measured effects cell line dependent?

Figure 1

The authors identify PRMT6 dependent splicing events and verify by rescue.

F1C: Please give numbers of genes within the pie chart

F1D: The bar graph could have its own labelling as Figure 1F. The error bars are huge and overlapping. Despite n=4 the significance is high. Is this realistic? Please recheck. If the variance is that high, does it still have biological meaning?

F1G: The bar graph could have its own labelling as Figure 1G

Figure 2

The authors examine a published data set for H3R2me2 at the alternatively spliced genes and conclude that PRMT6 mediated H3R2me is not predominantly involved in splicing regulation.

F2A: The GO terms are hardly readable. Please consider a different solution.

Figure 3

The authors relate splicing events occurring during neuronal differentiation of the cell line used with PRMT6 dependent splicing events. They identified 24 genes which are alternatively spliced during differentiation and are also PRMT6 dependent.

F3E: GO term analysis of differentiation dependent spliced genes was performed. However, is this relevant? The focus should be on the 24 common genes. The GO terms are hardly readable.

Figure 4

The authors repeated experiments of Bouchard et al and showed that pluripotency markers are altered during differentiation in loss of PRMT6 cells. They show that DNMT3 is alternatively spliced in the process.

F4C: The error bars show high deviation. Is this still biological relevant?

Minor points

Figure 1F: Western Blot of ...

Here the method is mentioned first. It would be more elegant to start with the purpose of the experiment.

Does PRMT6 bind DNA close to the alternative splice site? Are there PRMT6 ChIP data available?

Manuscript: LSA-2024-02946

We would like to thank the editor for considering and handling our manuscript and for inviting the submission of a revised manuscript addressing the Reviewers' comments. We are very grateful to the three reviewers for their helpful, valuable and thorough comments. As a result of the reviewers' suggestions, we believe that we could strengthen our findings in the revised manuscript. In the following we will address all issues raised by the reviewers point-by-point. In **italics/bold** are the original comments of the reviewers and below are our responses indicated in blue. All text modifications appear highlighted in yellow in the revised manuscript.

Response to the Reviewers:**Reviewer 1**Major comments:Point 1:

The complete list of splicing events related to Figure 1 and Figure 3 with details of splicing type and genomic coordinates should be provided as supplementary table.

We thank the reviewer for her/his comment and accordingly included the details of all the splicing changes $\Delta\text{PSI} \geq 0.05$ with a probability ≥ 0.9 (splicing type, genomic coordinates) in the new Table EV1.

Point 2:

For all semi-quantitative RT-PCRs, a loading control (e.g. ACTB or GAPDH) should be included and normalized when quantifying the amplicons. Sometimes, the band differences may arise from un-uniform loading of samples. For example, in Fig. 1D, the bands in KO sample seems brighter compared to CT.

We thank the reviewer for raising this point. As suggested by the reviewer, we have incorporated the corresponding PCR of *GAPDH* as reference/housekeeping gene in the agarose gel images of all semi-quantitative RT-PCRs of the revised manuscript (Fig. 1, Fig. 4, Fig. EV3, Fig. EV5; Fig. EV11). In case of *PRMT4* (Fig. 1D), the *GAPDH* RT-PCR indicates that the reference amplification is stronger in the CT compared to KO (not vice versa). We did not use the *GAPDH* signals for normalization in our quantifications, as we never directly compared the *PRMT6*-dependent ΔE signals of CT and KO conditions with each other, but we densitometrically quantified the relative abundance of the *PRMT6*-dependent splicing event with respect to the full-length isoform within the same condition. For example, in Fig. 1D we calculated the two signals of *PRMT4* FL and $\Delta 15$ to 100% in each individual condition (either CT or KO) and express how much the $\Delta 15$ isoform made up in % in each condition. This % of the $\Delta 15$ isoform (relative to the FL isoform) was the compared between the two conditions (CT/KO) and is displayed in the quantification (Fig. 1G).

Point 3:

The authors don't mention why they chose to validate PRMT4 and CHASERR. Are these the most significantly spliced?

We thank the reviewer for this question. In the revised manuscript we enlarged the number of validated splicing targets (additionally *FLNB* and *CALD1*, Figure 1F, I, M and EV3) and clarified in the revised manuscript text why we chose them. For validation, we selected gene transcripts

of the major PRMT6-regulated splicing categories (i.e. cassette exons and mutually exclusive exons), among them splicing events with high Δ PSI values (*PRMT4*, *CHASERR*, *CALD1*).

Point 4:

While the PRMT4 splicing levels in Fig. 1D as shown violin plots, more-or-less match with gel images, the CHASERR levels in E doesn't seem significantly different between CT and KO, at least visually. Could authors clarify this?

The reviewer is correct that there is a discrepancy between the violin plots in the *CHASERR* Voila analysis of the sequencing data and the semi-quantitative/standard RT-PCR results in the agarose gel electrophoresis, where the exon skipping vs. exon inclusion ratios do not fully represent the RNA-seq data (Fig. 1E, H). We observed such discrepancy between RNA-seq quantification and the semi-quantitative PCR validation for a few splicing targets (additionally to *CHASERR* also in case of *FLNB*, depicted in the new Fig. 1F, I). This can be due to different molecular biases in the two experimental procedures. In RNA-seq library preparation, RNA shearing, reverse transcription priming and linker ligations may introduce other biases compared to RNA isolation and the following different reverse transcription reactions that are performed prior to PCR. During semi-quantitative PCR, the composition and positioning of the primers, the longer amplicons as well as the GC-content of the region may sometimes alter PCR efficiency. Such standard PCR-specific parameters do normally not influence the short reads of NGS-based approaches or the relatively short amplicons of quantitative (q) PCR experiments. We have mentioned and explained this discrepancy in the revised manuscript text. Although the extend of some of our splicing effects is different when comparing the RNA-seq/qPCR data to the standard PCR results, the PRMT6- as well as differentiation-mediated splicing tendencies of the RNA-seq were verified by our validations, as also confirmed by the correlation scatter plot analysis of RNA-seq-derived and RT-PCR-derived Δ PSI values (Figure EV4).

Point 5

It would be great if authors show RT-PCR validation (gel images) of PRTM4 Ex4 splicing in their rescue experiments (related to Fig1F and G). It will also add strength to the rescue experiments if they include CHASERR Ex2 splicing rescue.

We acknowledge the reviewer's comment. As suggested by the reviewer, we performed the qPCR analysis also for *CHASERR* and an additional splicing target (*FLNB*) in the rescue cells. Wild type, but not mutant, PRMT6 reversed the change in alternative splicing of *CHASERR* and *FLNB* caused by PRMT6 deletion. These results have been incorporated in the revised manuscript (Fig. 1L, M). Unfortunately, we were not able to additionally validate the PRMT6 effects on splicing of *PRMT4*, *CHASERR* and *FLNB* by standard RT-PCR in the rescue experiment, even though we tried our best. The effects were only very moderately visible on agarose gel images and difficult to reproduce. Thus, we finally obtained validations of the *PRMT4*, *CHASERR* and *FLNB* splicing events in rescue experiments by RT-qPCR but not by semi-quantitative RT-PCR.

Point 6:

Authors mention "H3R2me2 deposition not overlapping with the splice site (e.g. PRMT4, Figure 2C) or even no H3R2me2a enrichment within in the target gene locus (e.g. CHASERR, Figure 2D)". It is very difficult to see whether the deposition overlaps or not at the splice site. Not sure what is the best way to show this, may be zooming in with high-magnification image of the splice site to compare? Is there a way to quantify the deposition for a given locus? If so, it would be very useful to include it.

We thank the reviewer for her/his question regarding the visualization of the H3R2me2a enrichment. In our previous report on PRMT6 and H3R2me2a (Bouchard et al., 2018), we quantified the ChIP-seq-derived H3R2me2a signal relative to the IgG control ChIP. H3R2me2a peaks are defined by an enrichment ≥ 2 -fold enrichment above IgG. Thus, we included in all genome browser views (Fig. 2C, D, Fig. EV7) of the revised manuscript the corresponding IgG ChIP data sets, which visualize the background enrichment of a given locus. For further clarification of the location of H3R2me2a peaks in the specific genomic regions, we highlighted the H3R2me2a peak with green arrows (below the H3R2me2a/CT tracks in Fig. 2C, D, Fig. EV7).

Point 7:

In the scatter plot presented in Fig. 3D, there are genes (e.g. CLSTN1, DNMT3B) are labelled more than once. Are these different splicing events within the same gene? If so, it would be useful if they are included in the validation alongside F and G.

The reviewer is correct that in Fig. 3D for some genes, such as *CLSTN1* and *DNMT3B*, multiple splicing events are shown. As suggested by the reviewer, we validated a second splicing event in *DNMT3* (E21/22) by RT-PCR analysis, which is included in the revised manuscript (Fig. EV11E). Unfortunately, we could not validate the *CLSTN1* splicing event due to failure of the PCR amplification despite immense efforts to improve the primer design.

Point 8:

Statistics in Fig. 4C is not confusing. In PRMT6 WT, the difference (Dox- and +; ATRA+ +), doesn't look significant at all, but it shown as **, could authors check this and other comparisons as well?

We thank the reviewer for bringing up this important and critical point to our attention. We checked again our statistical analysis (paired t-test/Student's t-test) of former Fig. 4C (in the revised manuscript Fig. 5C) and reproduced the significance values shown in the original manuscript. Given that the data derive from 5 biological replicates, we decided to visualize the associated data points of the respective replicate in the same color. This color-coded visualization demonstrates that the absolute values in % of the *DNMT3B* $\Delta E10$ isoform differ considerably between the 5 replicates. Importantly, the tendencies are significantly reproducible in the 5 biological replicates, i.e. re-expression of PRMT6 WT, but no PRMT6 MUT, increases the abundance of the differentiation-associated $\Delta E10$ isoform of *DNMT3B*. As suggested by the reviewer we checked the significance calculations also for the other analog comparisons (Fig. 1G-I and Fig. 4C, D – former Fig. 1D, E and Fig. 3F, G) in our manuscript and confirmed the significance values shown in our original manuscript. To clarify the affiliation of data points to the same biological replicate, we used also here the color-coding, similar to Fig. 5C (former Fig. 4C).

Point 9:

PRMT6's splicing-related mechanism is thought to involve methylating RNA binding proteins. So, it is reasonable to say that it would influence a set of RBP methylation to regulate splicing. Authors could analyze some putative RBPs in KO background to probe PRMT6's involvement. Without this the schematic presented in Fig. 4D is too vague.

The reviewer is correct that our model with respect to arginine methylation of RBPs by PRMT6 is vague and experimentally not substantiated. In unpublished affinity purification of PRMT6 from MCF7 cell extracts and mass spectrometry, we found that PRMT6 interacts with numerous RNA binding proteins and splicing factors, such as several HNRNP proteins,

NF45/ILF2 and NF90/ILF3. Independent of our observation, NF45 has been reported to interact with PRMT6 and to be methylated in a PRMT6-dependent manner (Avasarala 2020, Schneider 2021; Wu iScience 2024). Moreover, NF45 and NF90 have recently been shown to regulate splicing of cassette exons and mutually exclusive exons (Haque et al., 2023). To address the reviewer's comment and to strengthen our hypothesis that PRMT6 might influence splicing by arginine methylation of RBPs and splicing modulators, we analyzed a potential interaction of NF45 as well as NF90 with PRMT6 in NT2/D1 cell lysates by co-IP / Western blot experiments. This analysis revealed that PRMT6 interacts with NF45, but not with NF90 in NT2/D1 cells. To address whether NF45 would be also a substrate of PRMT6 in NT2/D1 cells, we performed immunoprecipitation of NF45 from NT2/D1 CT and KO cell lysates followed by anti-ADMA Western blot analysis indicating that NF45 is asymmetrically di-methylated in a PRMT6-dependent manner. These novel findings are now included in Figure 5D and E of the revised manuscript. Although these findings do not show that PRMT6 affects cassette exons via interaction and modification of NF45, we think that these additional data might allow us to hypothesize in our final model on such a potential mechanism.

Minor comments:

Point 1:

The authors have performed RNA-seq analysis to reveal alternative splicing using their previously published datasets (Bouchard et al 2018). However, they don't mention in both 2018 and current paper about the read depth (how many million reads). This information is crucial as the read depth influences alternative splicing analysis outcome. It would be great if authors provide this information in the methods.

We thank the reviewer for her/his attentive comment. Our originally in Bouchard et al. (2018) published RNA-seq of CT/KO NT2/D1 cells +/- ATRA (in triplicates, i.e. 12 samples) is publicly available at GEO (accession number GSE107612). The sequencing depth is on average 48 million reads. The exact numbers of input reads and of uniquely mapped reads of the 12 samples has been included in Fig. EV1A of the revised manuscript.

Point 2:

Full uncropped raw images of agarose gel and western blot images should be provided as supplementary images.

We acknowledge the reviewer's comment and have compiled original source data files with uncropped/unprocessed agarose gel images and Western blot data images for the main figures (Fig. 1, Fig. 3, Fig. 5) and the EV figures (Fig. EV3, Fig EV5, Fig. EV11) of the manuscript.

Point 3:

The title is very generic and might mislead. It could mean pluripotency in ES cells in mouse, human etc. As all the experiments in this paper were done on NT2/D1 cell line (which is not bad), the authors could consider including it in the title and revise it to precisely reflect the work.

We thank the reviewer for bringing up this important issue about the manuscript title to our attention. We agree with the reviewer and have rephrased our title as follows: Assessment of PRMT6-dependent functions in alternative splicing of pluripotent and differentiating NT2/D1 cells.

Reviewer 2**Major comments****Point 1:**

To strengthen the data, the authors should provide additional validation using RT-PCR and agarose gel assays beyond the two events shown in Figures 1 and 3. Ideally, profiling 10-20 splicing events, providing the corresponding agarose gels in the supplementary materials, and generating correlation scatter plots between RNA-Seq-derived and RT-PCR-derived PSI values would substantially bolster the validity of the findings.

We thank the reviewer for her/his attentive and valuable comment. As suggested by the reviewer we selected additional splicing targets of our RNA-seq analysis for validation by semi-quantitative RT-PCR. This allowed us to validate an additional 8 splicing events, which have been incorporated in our revised manuscript in Fig. 1F, Fig. EV3 and Fig. EV11 including also the corresponding agarose gel images. Together with the 4 validated events, which were already presented in the original manuscript, we thus validated in total 12 splicing events. Furthermore, as suggested by the reviewer, we generated a correlation scatter plot analysis of the RNA-seq-derived and RT-PCR-derived Δ PSI values (Figure EV4) revealing that the PRMT6- as well as the differentiation-mediated splicing alterations of the RNA-seq show a good correlation with our RT-PCR validations ($r=0.84$, $p<0.0001$).

Point 2:

The study offers limited mechanistic insights into how PRMT6 affects alternative splicing. While identifying a specific PRMT6-targeted splicing factor might be beyond the scope of this study, the authors should at least attempt to determine whether the observed splicing changes are a direct result of PRMT6's methyltransferase activity or if they are secondary to altered differentiation status in PRMT6 knockout cells.

We thank the reviewer for bringing up this important points to our attention. To examine whether regulation of alternative splicing by PRMT6 is dependent on its enzymatic activity, we performed rescue experiments in NT2/D1 KO cells, which we had already included in our original manuscript and which we even extended in the revised manuscript. To this end, we re-expressed wild type (WT) or enzymatically inactive mutant (MUT) PRMT6 in a doxycycline-inducible manner in KO cells, as verified on RNA and protein levels (Figure EV6, Figure 1J). Wild type, but not mutant, PRMT6 reversed the changes in alternative splicing caused by PRMT6 deletion of *PRMT4* (Fig. 1K) *CHASERR* (Fig. 1L), *FLNB* (Fig. 1M) and *DNMT3B E10* (Fig. 5C).

Moreover, to gain some mechanistic insights into how PRMT6 might affect alternative splicing, we revisited published interactome studies showing that PRMT6 interacts with numerous RBPs and SFs, such as several HNRNP proteins, NF45/ILF2 and NF90/ILF3 (Wei et al, 2021; Avasarala et al, 2020; Schneider et al, 2021). Interestingly, NF45 has recently been reported to be methylated in a PRMT6-dependent manner and to contribute to alternative splicing of cassette exons and mutually exclusive exons (Wu et al, 2024; Haque et al, 2023), which are the predominant splicing effects of PRMT6 in NT2/D1 cells. To address whether PRMT6 and NF45 might also associate with each other in NT2/D1 cells, we performed co-immunoprecipitation analysis and found that the two proteins interact at the endogenous level in NT2/D1 cells. To determine whether NF45 would also serve as a substrate of PRMT6 in NT2/D1 cells, we performed immunoprecipitation of NF45 from NT2/D1 CT and KO cell lysates followed by Western blot analysis for asymmetrically di-methylated arginine (ADMA). Our results show that NF45 is stained for ADMA in CT lysates, whereas the staining is diminished in the KO condition, indicating that NF45 is methylated in a PRMT6-dependent manner in

NT2/D1 cells. These new findings have been incorporated in the revised manuscript (Fig. 5D and E) and suggest that PRMT6 might potentially regulate cassette exons and mutually exclusive exons via interaction and modification of splicing modulators, such as NF45, in NT2/D1 cells.

Point 3:

The manuscript does not adequately address how PRMT6 affects NT2/D1 cell differentiation. To clarify this, the authors should expand Figure 4A-B to include wild-type (WT) cells, not just the knockout (KO) cells. Additionally, monitoring a neuronal differentiation marker, such as TUBB3, would provide further clarity on PRMT6's role in differentiation.

We thank the reviewer for her/his critical comment. NT2/D1 cells derive from human embryonal carcinoma cells and were originally established by Lee and Andrews (1986). These cells have been intensively studied in the past, including by our lab, and have been shown to differentiate into neuronal cells upon treatment with ATRA. This neuronal differentiation is accompanied by the downregulation of pluripotency genes, such as *OCT4*, *NANOG* and *SOX2*, and the upregulation of neurogenesis-related genes, such as the rostral *HOXA* gene cluster, *RARB*, *MEIS1* and *MEIS2*. Although we could not detect a transcriptional change of *TUBB3* in the course of neuronal differentiation of NT2/D1 cells, as suggested by the reviewer, we included a more extensive characterization of the differentiation-associated transcriptional reprogramming in NT2/D1 PRMT6 wild type (CT) cells in Fig. EV8. Given that our previous work (Hyllus et al, 2007; Stein et al, 2016; Bouchard et al, 2018) revealed that PRMT6 is an important transcriptional coregulator of the gene expression program in NT2/D1 cells and contributes to their ATRA-induced neuronal differentiation by regulating transcriptional initiation of pluripotency (e.g. *Oct4*) and differentiation genes (e.g. the *HOXA* gene cluster), we analyzed in the present manuscript whether alternative splicing associates with neuronal differentiation in NT2/D1 cells and whether this differentiation-associated splicing would be impacted by PRMT6.

Minor comments

Point 4:

Gene expression tracks should be included in Figures 4C-D and supplementary Figures EV4A-C to provide a more comprehensive view of the data.

We thank the reviewer for her/his insightful comment and have included the RNA-seq tracks in Fig. 2C, D and Fig. EV7A-C (former Fig. EV4A-C), as suggested by the reviewer.

Point 5:

The ribosomal proteins highlighted in Extended Data Figure EV4 could be among the genes validated by RT-PCR assays, as suggested in the major comment 1.

We thank the reviewer for attracting our attention to the alternatively spliced transcripts of ribosomal proteins. Unfortunately, we were not able to additionally validate the PRMT6 effects for the alternative splicing events of ribosomal genes by standard RT-PCR, even though we tried our best. We failed to establish the PCR amplification for these genes despite immense efforts to improve the primer design and the PCR conditions.

Point 6:

A representative agarose gel corresponding to Figure 4C could be included either as a supplementary figure or positioned above the bar plot for clearer visualization.

We thank the reviewer for her/his suggestion. However, the former Fig. 4C (Fig. 5C in the revised manuscript) is a RT-qPCR experiment for which we do not have a corresponding agarose gel image.

Reviewer 3

Major comments

Point 1:

The mechanisms how PRMT6 influences splicing is elusive. This limits the impact of the data.

We thank the reviewer for her/his critical remark. Our data excluded a few mechanisms for the majority of PRMT6-regulated splicing events in NT2/D1 cells, such as direct coupling of splicing and transcriptional regulation, the influence of H3R2me2a or an accompanying SACS (at least with regard to the so far studied histone marks). Nevertheless, we attempted to gain at least some more mechanistic insights into how PRMT6 might affect alternative splicing during the revision process. Therefore, we revisited published interactome studies showing that PRMT6 interacts with numerous RBPs and SFs, such as several HNRNP proteins, NF45/ILF2 and NF90/ILF3 (Wei et al, 2021; Avasarala et al, 2020; Schneider et al, 2021). Interestingly, NF45 has recently been reported to be methylated in a PRMT6-dependent manner and to contribute to alternative splicing of cassette exons and mutually exclusive exons (Wu et al, 2024; Haque et al, 2023), which are the predominant splicing effects of PRMT6 in NT2/D1 cells. To address whether PRMT6 and NF45 might also associate with each other in NT2/D1 cells, we performed co-immunoprecipitation analysis and found that the two proteins interact at the endogenous level in NT2/D1 cells. To determine whether NF45 would also serve as a substrate of PRMT6 in NT2/D1 cells, we performed immunoprecipitation of NF45 from NT2/D1 CT and KO cell lysates followed by Western blot analysis for asymmetrically di-methylated arginine (ADMA). Our results show that NF45 is stained for ADMA in CT lysates, whereas the staining is diminished in the KO condition, indicating that NF45 is methylated in a PRMT6-dependent manner in NT2/D1 cells. These new findings have been incorporated in the revised manuscript (Fig. 5D and E) and suggest that PRMT6 might potentially regulate cassette exons and mutually exclusive exons via interaction and modification of splicing modulators, such as NF45, in NT2/D1 cells.

Point 2:

It is not clear if differential PRMT6 dependent splicing contributes to neuronal differentiation. Does the alternatively spliced DNMT contribute?

We thank the reviewer for this question. As we discuss in our manuscript, DNMT3B $\Delta E10$ is the predominant splice isoform in ATRA-treated NT2/D1 cells (Fig. 4B, D). This differentiation-induced splicing event is PRMT6-dependent (Fig. 4A, C, Fig. 5C) and has been reported to coincide with a loss of pluripotency and a gain of differentiation (Gopalakrishna-Pillai & Iverson, 2011), suggesting that PRMT6 might contribute through splicing regulation of DNMT3B to the differentiation process.

Point 3:

The title mentions an effect of alternative PRMT6 dependent splicing on pluripotency. Do the data really support this notion?

We thank the reviewer for bringing up this critical question about the manuscript title to our attention. We agree with the reviewer and have rephrased our title as follows: Assessment of PRMT6-dependent functions in alternative splicing of pluripotent and differentiating NT2/D1 cells.

Point 4:

Mostly available data are reanalyzed. Which is fine. However, are the measured effects cell line dependent?

We thank the reviewer for her/his interesting question. To examine whether the PRMT6-regulated splicing events are general or cell line-dependent effects, for example specific for NT2/D1 cells, we generated HeLa, HEK293T and U2OS cell lines containing a PRMT6 knockout and analyzed the corresponding short and long isoforms of *PRMT4* and *FLNB* by standard RT-PCR. The three cell lines displayed in their PRMT6 wild type state (CT) a more or less predominant expression of the PRMT4 $\Delta E15$ and FLNB $\Delta E2$ isoform in contrast to NT2/D1 CT cells, in which the FL isoforms prevail for both transcripts. PRMT6 deficiency (KO) did not affect the PRMT4 isoform ratio in HeLa cells compared to CT cells, but caused decreased $\Delta E15$ levels in HEK293 cells and increased $\Delta E15$ levels in U2OS cells, of which the latter effect is similar to the PRMT6-mediated regulation in NT2/D1 cells. PRMT6 deletion did not alter the *FLNB* isoform expression in the three tested cell lines. These findings have been incorporated in the revised manuscript (Fig. EV5) and suggest that the splicing targets of PRMT6, as examined here for two examples, are regulated in a cell line-dependent manner.

Points to Figure 1:

The authors identify PRMT6 dependent splicing events and verify by rescue.

F1C: Please give numbers of genes within the pie chart

F1D: The bar graph could have its own labelling as Figure 1F. The error bars are huge and overlapping. Despite n=4 the significance is high. Is this realistic? Please recheck. If the variance is that high, does it still have biological meaning?

F1G: The bar graph could have its own labelling as Figure 1G

We thank the reviewer for her/his valid suggestions. We included the numbers in the pie chart of Fig. 1C. All bar graphs of Fig. 1 obtained their own labelling in the revised manuscript. We checked again our statistical analysis (paired t-test/Student's t-test) of former Fig. 1D (in the revised manuscript Fig. 1G) and reproduced the significance values shown in the original manuscript. Given that the data derive from 4 biological replicates, we decided to visualize the associated data points of the respective replicate in the same color. This color-coded visualization demonstrates that the absolute values in % of the *PRMT4* $\Delta E15$ isoform differ considerably between the 4 replicates. Importantly, the tendencies are significantly reproducible in the 4 biological replicates. We checked the significance calculations also for the other analog comparisons (Fig. 1H, I and Fig. 4C, D) in our manuscript and confirmed the significance values shown in our original manuscript. To clarify the affiliation of data points to the same biological replicate, we used in all these analyses the color-coding, similar to Fig. 1G.

Points to Figure 2:

The authors examine a published data set for H3R2me2 at the alternatively spliced genes and conclude that PRMT6 mediated H3R2me is not predominantly involved in splicing regulation.

F2A: The GO terms are hardly readable. Please consider a different solution.

We thank the reviewer for this helpful indication and enlarged the terms of the GO analysis.

Points to Figure 3:

The authors relate splicing events occurring during neuronal differentiation of the cell line used with PRMT6 dependent splicing events. They identified 24 genes which are alternatively spliced during differentiation and are also PRMT6 dependent.

F3E: GO term analysis of differentiation dependent spliced genes was performed. However, is this relevant? The focus should be on the 24 common genes. The GO terms are hardly readable.

We thank the reviewer for this helpful suggestion and enlarged the terms of the GO analysis. We also performed GO analysis with the 24 common (differentiation- and PRMT6-dependent) alternative splicing events. Unfortunately, this analysis did not result in any significant GO terms due to the small sample size.

Points to Figure 4:

The authors repeated experiments of Bouchard et al and showed that pluripotency markers are altered during differentiation in loss of PRMT6 cells. They show that DNMT3 is alternatively spliced in the process.

F4C: The error bars show high deviation. Is this still biological relevant?

We thank the reviewer for bringing up this important and critical point to our attention. We checked again our statistical analysis (paired t-test/Student's t-test) of former Fig. 4C (in the revised manuscript Fig. 5C) and reproduced the significance values shown in the original manuscript. Given that the data derive from 5 biological replicates, we decided to visualize the associated data points of the respective replicate in the same color. This color-coded visualization demonstrates that the absolute values in % of the *DNMT3B* Δ E10 isoform differ considerably between the 5 replicates. Importantly, the tendencies are significantly reproducible in the 5 biological replicates, i.e. re-expression of PRMT6 WT, but no PRMT6 MUT, increases the abundance of the differentiation-associated Δ E10 isoform of *DNMT3B*. Therefore, we think that the results are biological relevant.

Minor comments

Point 1:

Figure 1F: Western Blot of ...

Here the method is mentioned first. It would be more elegant to start with the purpose of the experiment.

We thank the reviewer for this suggestion and have rephrased the figure legend.

Point 2:

Does PRMT6 bind DNA close to the alternative splice site? Are there PRMT6 ChIP data available?

We thank the reviewer for bringing up this critical point. However, we did not succeed in generating PRMT6 ChIP-seq analyses and are not aware of any other study reporting genome-wide mapping of PRMT6 chromatin binding sites. Therefore, we cannot answer the reviewer's question.

January 13, 2025

RE: Life Science Alliance Manuscript #LSA-2024-02946-TR

Prof. Uta-Maria Bauer
Philipps University of Marburg
Institute for Molecular Biology and Tumor Research (IMT)
Hans-Meerwein-Str. 2
Marburg 35043
Germany

Dear Dr. Bauer,

Thank you for submitting your revised manuscript entitled "Assessment of PRMT6-dependent alternative splicing of pluripotent and differentiating NT2/D1 cells". We would be happy to publish your paper in Life Science Alliance pending final revisions necessary to meet our formatting guidelines.

- please address Reviewer 1's remaining comment
- please be sure that the authorship listing and order is correct
- LSA allows supplementary figures, but no EV Figures; please update your callouts for the Supplementary Figures in the manuscript Fig EV1A=Fig S1A, while supplementary figures use the system supplementary Fig S1..the same applies to the tables
- please add Keywords for your manuscript to our system
- please add the Twitter handle of your host institute/organization as well as your own or/and one of the authors in our system
- please note that the titles in the system and manuscript file must match
- please use the [10 author names et al.] format in your references (i.e., limit the author names to the first 10)
- please add callouts for Figure 11A-F to your main manuscript text
- you may want to consider uploading Figure 6 as a Graphical Abstract rather than as a figure, but this it up to you

LSA now encourages authors to provide a 30-60 second video where the study is briefly explained. We will use these videos on social media to promote the published paper and the presenting author (for examples, see <https://docs.google.com/document/d/1-UWCfbE4pGcDdcgzcmiuJl2XMBJnxKYeqRvLLrLSo8s/edit?usp=sharing>). Corresponding or first-authors are welcome to submit the video. Please submit only one video per manuscript. The video can be emailed to contact@life-science-alliance.org

A. FINAL FILES:

B. MANUSCRIPT ORGANIZATION AND FORMATTING:

Sincerely,

Reviewer #1 (Comments to the Authors (Required)):

The revised version of the manuscript by Eudenbach et al., is significantly improved. The authors have addressed all of the comments raised to improve the manuscript.

A minor comment to note: the authors have used same GAPDH gel images more than once (Ex, Fig 1E,F; Fig 4A, C; EV11). It is understandable that same control PCR was used for validating different splicing events. This is absolutely fine. However, to avoid any unnecessary post-publication debate about image duplication, the authors could consider combining the gel image panels together (for ex. Fig 1E and F) and use single GAPDH image.

Reviewer #2 (Comments to the Authors (Required)):

The authors have addressed all my concerns and I support the publication of this manuscript in Life Science Alliance.

Reviewer #3 (Comments to the Authors (Required)):

Dear Editor, Dear Authors,

Considering the response of the authors to the reviewer questions, all my remaining concerns have been addressed.

Kind regards

Manuscript: LSA-2024-02946-TRR

Dear Dr. Sawey,

Thank you very much for considering and handling our manuscript entitled "Assessment of PRMT6-dependent alternative splicing in pluripotent and differentiating NT2/D1 cells" and for giving us the opportunity to submit a final revision.

We are very grateful to the reviewers for their very helpful comments during the review process and their positive statements on our revised manuscript.

We agree with the concern of reviewer 1 and have followed her/his advice to remove all duplicates of the GAPDH control PCR gel images in the relevant figures. Thank you for bringing this matter to our attention. In the following we will address the issue raised by the reviewer 1. In italics/bold is the original comment of the reviewer and below is our response indicated in blue. We hope very much that these changes will be approved by the reviewer.

Thank you very much in advance for dealing again with our manuscript.

Sincerely,

Uta-Maria Bauer

Response to Reviewer 1:**Comment:**

A minor comment to note: the authors have used same GAPDH gel images more than once (Ex, Fig 1E,F; Fig 4A, C; EV11). It is understandable that same control PCR was used for validating different splicing events. This is absolutely fine. However, to avoid any unnecessary post-publication debate about image duplication, the authors could consider combining the gel image panels together (for ex. Fig 1E and F) and use single GAPDH image.

We thank the reviewer for bringing this matter to our attention. We fully agree with the concern of the reviewer and have followed her/his advice to remove all duplicates of the GAPDH control PCR gel images in the figures 1F, 4C und S11C-F. Instead of displaying duplicates, the GAPDH PCR gel images are exhibited once in the figure panels 1E, 4A, S11A, B, and it is stated in the relevant subsequent figure panels (indicated with an empty box) and in the corresponding figure legends that the control PCR is identical in Fig. 1F (to 1E), 4C (to 4A), S11C, D and F (to S11A), as well as in S11E (to S11B). We hope very much that these changes will be approved by the reviewer.

January 21, 2025

RE: Life Science Alliance Manuscript #LSA-2024-02946-TRR

Prof. Uta-Maria Bauer
Philipps University of Marburg
Institute for Molecular Biology and Tumor Research (IMT)
Hans-Meerwein-Str. 2
Marburg 35043
Germany

Dear Dr. Bauer,

Thank you for submitting your Research Article entitled "Assessment of PRMT6-dependent alternative splicing in pluripotent and differentiating NT2/D1 cells". It is a pleasure to let you know that your manuscript is now accepted for publication in Life Science Alliance. Congratulations on this interesting work.

DISTRIBUTION OF MATERIALS:

Again, congratulations on a very nice paper. I hope you found the review process to be constructive and are pleased with how the manuscript was handled editorially. We look forward to future exciting submissions from your lab.

Sincerely,
